# Sources of confidence in value-based choice

Jeroen Brus [1,2✉], Helena Aebersold [3], Marcus Grueschow [4] & Rafael Polania [1,2✉]

Confidence, the subjective estimate of decision quality, is a cognitive process necessary for learning from mistakes and guiding future actions. The origins of confidence judgments resulting from economic decisions remain unclear. We devise a task and computational framework that allowed us to formally tease apart the impact of various sources of confidence in value-based decisions, such as uncertainty emerging from encoding and decoding operations, as well as the interplay between gaze-shift dynamics and attentional effort. In line with canonical decision theories, trial-to-trial fluctuations in the precision of value encoding impact economic choice consistency. However, this uncertainty has no influence on confidence reports. Instead, confidence is associated with endogenous attentional effort towards choice alternatives and down-stream noise in the comparison process. These findings provide an explanation for confidence (miss)attributions in value-guided behaviour, suggesting mechanistic influences of endogenous attentional states for guiding decisions and metacognitive awareness of choice certainty.

[1] Decision Neuroscience Lab, Department of Health Sciences and Technology, ETH Zurich, Zurich, Switzerland. [2] Neuroscience Center Zurich, Zurich, Switzerland. [3] Epidemiology, Biostatistics and Prevention Institute, University of Zurich, Zurich, Switzerland. [4] Zurich Center for Neuroeconomics (ZNE), Department of Economics, University of Zurich, Zurich, Switzerland. ✉email: jeroen.brus@hest.ethz.ch; rafael.polania@hest.ethz.ch

The ability to evaluate the quality of our own decisions in the absence of immediate feedback is a fundamental aspect of cognition, which can be used to revise decisions and guide future behavior. This kind of subjective evaluation of choice is known as confidence, which is particularly relevant in the domain of economic decisions, where subjective evaluations of choice can have far-reaching implications for the welfare of individuals and social groups[1–3]. For instance, after finally deciding on a dish in a restaurant, we might feel that it was actually not the dish we wanted to eat, potentially leading us to revise our meal choice[4]. Likewise, we may introspect decisions about what house we opted to buy, or what financial investment we have made. Despite its importance, little is known about the mechanisms underlying confidence originating from subjective value-based decisions.

Arguably, most of our knowledge regarding confidence mechanisms originates from the domain of perceptual decision making[1,5–10]. A potential reason is that, in purely perceptual decision tasks, experimenters have full control of the objective stream of evidence presented to the decision-maker, thus allowing to examine how each objective ingredient of the input stimuli affects the choice and the resulting confidence evaluation.

Normative approaches formulate that confidence reflects an optimal estimate that the decision was correct[11,12] and is a direct transformation of evidence strength[13–15]. This rationale is particularly relevant to one of the central arguments that have sparked research in the last years: whether confidence represents an accurate index of decision uncertainty[11,16,17]. Interestingly, earlier studies found that people indeed rely on the strength of evidence, but do not directly consider the uncertainty on that evidence when they make confidence reports[18], nor do they necessarily use the evidence available against their choice[19,20]. All these studies aimed at dissecting the relationship between choice behavior and its resulting confidence evaluations[20–24]. However, this has not been formally studied in the domain of value-based choices.

A potential reason for this gap is that in the domain of value-based choices neither the experimenter nor the decision-maker has complete access to the subjective values of the alternatives comprising a decision, where values are potentially sampled from multiple attributes stored in memory[25]. This is critical, as an interpretation of confidence from decisions that rely on subjective-value estimations run the risk of being misattributed to aspects of the choice process that are actually choice-irrelevant. Therefore, it is important to establish whether processes that lead to confidence reports in value-based choice overlap or differ from those supporting simple perceptual decisions. Recent developments in the study of subjective valuation that more formally take into consideration some of these limiting aspects might be promising approaches to elucidate the sources of confidence in value-based choices[26,27]. For instance, using some of these developments it might be possible to investigate how post-decision confidence reports are influenced by distinct sources of variability in the decision process while taking into account the distribution of individual preferences in a given environment or context.

Another key aspect to consider is that value-based choice tasks usually entail two or more alternatives for choice situated at different spatial locations of the visual field. That is, decision-makers foveate—often repeatedly via changes in eye-fixation—to one of the choice options at a time, thereby gathering evidence from each alternative[28]. Seminal studies in the attention literature clearly indicate substantial influences of reward value in biasing attention with direct consequences on choice processes[29–31]. Thus, an unresolved question is whether observers have the capability to introspect about their endogenous attentional states of the decision processes and whether these internal signals are used to inform post-decision confidence reports.

We argue that these issues have been difficult to tackle given the lack of mechanistic and formal models of decision behavior capable of dissecting the different components of choice processes that rely entirely on subjective value evaluations. Here, we address these issues via a combination of a behavioral task with eye-tracking and computational modeling, which allowed us to dissect what aspects of the choice process are linked to post-decision confidence reports. More specifically, we study whether confidence reports emerge from sources of uncertainty emerging during encoding and decoding operations of the choice alternatives, as well as how the interplay between gaze-shift dynamics, attentional and cognitive effort influence metacognitive awareness of choice certainty. Across two independent datasets, we find that human participants incorporate knowledge of trial-to-trial fluctuations in attentional effort in their confidence reports, thus revealing that contrary to standard specifications[28] attentional effort is highly dynamic across trials and deeply influences both choices and confidence. Further investigating the role of different forms of noise in the decision process, we find that trial-to-trial fluctuations in encoding noise do not influence confidence, while down-stream comparison noise does.

## Results
**Experiment**. We implemented a behavioral paradigm that allows teasing apart distinct sources of variability in decisions based on subjective values of the choice alternatives[26]. In the first part of the experiment, participants ($n = 33$) were presented with single food items and asked to indicate on a continuous rating scale their desirability to consume the presented item at the end of the experiment, (rating phase 1, Fig. 1a). Participants then rated the same items a second time (rating phase 2, Methods). Crucially, participants were not informed before rating phase 1 that a second rating phase would take place. This was important as it prevented participants from actively memorizing the location of the rating in the slider in the first phase, thus providing us with a more accurate measure of the variability in the value estimates. This procedure allows us to study whether trial-to-trial fluctuations in subjective value estimations are reflected on value coding/decoding operations rather than just random noise, and how this variability affects the quality of the decisions[26]. Crucially, this also allows studying whether this source of variation is directly related to post-decision confidence reports. To investigate this, the same participants underwent a series of incentive-compatible choices in which they selected from pairs of the previously rated food items the one item they preferred to eat (Fig. 1a). Given that one of the main goals of this study is to investigate the influence of visual fixation dynamics and attentional effort on decision behavior and confidence evaluations, we tracked the participant's eyes while performing incentive-compatible choices (Methods). Summary statistics of the usage of the preference and confidence scale, as well as summary statistics of visual fixations, are reported in Supplementary Figs. 1 and 2.

**Analyses and modeling roadmap**. In the following, we will present several analysis and modeling approaches, all with the goal of elucidating the influence of different components of the decision process on confidence reports. Here we briefly outline the different modeling approaches adopted in this work.

In the first part, we perform a "model-free" analysis investigating the influence of key decision variables on choice and confidence reports, such as VD and total value (TV) of the input alternatives, alongside the influence of value estimation variability obtained from the rating phases. The impact of these factors is separately studied in both choice consistency and confidence reports. These analyses will provide initial hints about

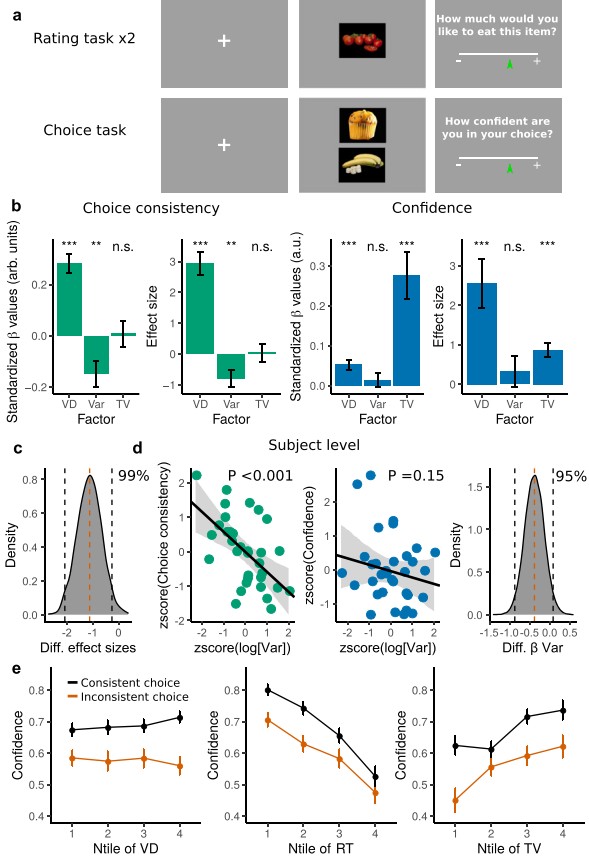

**Fig. 1 Experiment and regression analysis of choice consistency and confidence. a** Example display of the rating and choice task. Participants rated their desirability to eat the displayed food item. Next participants were asked to indicate which of the two food items they preferred to consume. After their choice, participants were asked how confident they were about their decision. **b** Standardized estimates of a multiple logistic regression on choice consistency (green) show that higher value difference (VD) leads to more consistent choices ($\beta = 0.28 \pm 0.04$, $P < 0.001$). Higher variability (Var) in the rating of the two alternatives leads to less consistent choices ($\beta = -0.15 \pm 0.05$, $P = 0.002$). The total value (TV) of the two items had no reliable influence on choice consistency ($\beta = 8.7 \times 10^{-3} \pm 0.05$, $P = 0.43$). Standardized estimates of a multiple linear regression on confidence reports show that higher VD lead to more confidence ($\beta = 0.05 \pm 0.01$, $P < 0.001$). Crucially, higher variability in the rating of the two alternatives does not have a reliable effect on confidence ($\beta = 0.01 \pm 0.02$, $P = 0.2$). Higher TV increases confidence ($\beta = 0.28 \pm 0.06$, $P < 0.001$). Error bars indicate the mean standard deviation of the posterior estimates. **c** The difference of the effect size of the influence of variability on choice consistency and confidence is significant with $P < 0.01$. Vertical red dashed line indicates the median, black vertical lines indicate the 95% highest density interval. **d** Participant's average level of variability in the rating task had a negative influence on average choice consistency of that participant ($\beta = -0.58 \pm 0.16$, $P < 0.001$, $r = -0.53$), however, this effect is not present for the same analyses performed on confidence reports ($\beta = -0.19 \pm 0.18$, $P = 0.15$, $r = -0.22$), $P$-values are based on the highest density interval of the posterior estimates. Gray shaded areas indicate the 95% confidence bands. The difference of the effect of average variability on choice consistency and confidence ratings is significant ($\Delta\beta_{cons-conf} = -0.39 \pm 0.24$, $P = 0.05$). **e** Confidence as a function of absolute value difference shows the qualitative signatures of confidence reports guided by its statistical definition. Confidence as a function reaction time shows signatures reported in previous work. Confidence as a function of total value confirms the quantitative results presented in panel (**b**). Data are presented as mean values ± SEM. For the whole figure $n = 33$ independent participants. Source data are provided as a Source Data file.

the degree of similarity with which these decision variables impact choices and confidence.

In the second part, we study dynamical aspects of the decision process on confidence reports by jointly incorporating choices, reaction times (RTs), and fixation patterns. This will be formally studied based on sequential sampling models which will allow dissecting the influence of latent variables of the decision process such as the degree of attentional effort and the decision evidence gain on confidence reports.

In the third part, we implemented a modeling approach that does not make use of RT information to investigate the influence of the same latent variables studied in the second part. This approach is appealing due to the following reasons: first, it is less computationally demanding. Second, access to RT data is not always possible in studies of perceptual and economic behavior. Third, as we will show in the last part of this article, this approach can be parsimoniously extended to formally incorporate the statistics of the environment, which in turn allows disentangling the influence of noise on confidence at different stages of the decision process such as the value encoding and downstream comparison processes.

**Value variability, choices, and confidence.** As commonly adopted in studies of value-based choices[26,28,32], here we define a consistent choice as a trial in which the subject choses the item they had assigned a higher average rating across the two previous ratings. Using a multi-factor hierarchical logistic regression, we found that choice consistency was influenced by the value difference (VD) between the two items' prior ratings: the higher the VD, the more consistent the choices ($\beta = 0.28 \pm 0.04$, $P < 0.001$; Fig. 1b), a result that is in line with previous work[4,26,28,33–36]. Importantly, choice consistency also depended on the variability in the value ratings: the higher the rating variability for the items on a given trial, the less consistent the decision ($\beta = -0.15 \pm 0.05$, $P = 0.002$; Fig. 1b), a result that replicates our previous work[26]. After controlling for VD and rating variability, we found that the TV of the alternatives in each trial had no influence on choice consistency ($\beta = 8.7 \times 10^{-3} \pm 0.05$, $P = 0.43$; Fig. 1b), a finding that is generally consistent with previous work[26,36]. We tested whether the null model (Choice ~ VD + Var; which in this case excludes TV) is more credible than the full model (Choice ~ VD + Var + TV) via the estimation of the Bayes Factor ($BF_{01}$). In this model comparison, we found $BF_{01} = 54$, which indicates "very strong" evidence[37] for the absence of TV effects on choices.

A key aspect of this study is to test whether trial-to-trial fluctuations in the estimation of reward value have an impact on value-based decisions. Specifically, canonical theories of confidence predict that these trial-to-trial fluctuations impact the confidence reports because this variation directly affects evidence for choice. Using a multi-factor regression we found, as expected, that VD has an impact on confidence reports ($\beta = 0.05 \pm 0.01$, $P < 0.001$; Fig. 1b). However and surprisingly, trial-to-trial fluctuations in the estimation of reward values had no impact on confidence reports ($\beta = 0.01 \pm 0.02$, $P = 0.2$; Fig. 1b). Again, we tested whether the null model (Confidence ~ VD + TV; which in this case excludes Var) is more credible than the full model (Confidence ~ VD + Var + TV). We found $BF_{01} = 450$, which indicates "very strong" evidence for the absence of rating variability effects on confidence. Furthermore, we found a significant difference in the effect sizes of the regressor capturing trial-to-trial fluctuations of subjective valuation between the choice and confidence models ($\Delta\beta_{EffectSize} = -1.11 \pm 0.48$, $P = 0.01$; Fig. 1c), indicating specificity of the effect on choice consistency but not on confidence. Extending this trial-to-trial effect of rating variability, we observed that each participant's

average level of variability in the rating task had a negative influence on average choice consistency of that participant ($\beta = -0.58 \pm 0.16$, $P < 0.001$, $r = -0.53$; Fig. 1d), and this effect was once again not present for the same analyses performed on confidence reports ($\beta = -0.19 \pm 0.18$, $P = 0.15$, $r = -0.22$; Fig. 1d). The difference of the effect of average variability on choice consistency and confidence ratings is marginally significant ($\Delta\beta_{cons-conf} = -0.39 \pm 0.24$, $P = 0.049$; Fig. 1d), however, this effect becomes more pronounced when removing two participants with outlying confidence rating behavior ($\Delta\beta_{cons-conf} = -0.53 \pm 0.25$, $P = 0.02$; Fig. 1d and Supplementary Figs. 1 and 3). Moreover, contrary to the findings in the choice consistency logistic regression, our data reveal that TV has a large positive impact on confidence reports ($\beta = 0.28 \pm 0.06$, $P < 0.001$; Fig. 1b). Once again, we found a significant difference in the effect sizes of the regressor capturing trial-to-trial fluctuations of subjective valuation between the choice consistency and confidence models ($\Delta\beta_{EffectSize} = -0.82 \pm 0.35$, $P = 0.009$), but this time confirming the specificity of the TV effect on confidence but not on choice consistency.

We investigated whether our data would still qualitatively capture the canonical signatures of confidence[16,38] despite these surprising results (Fig. 1b–d). Indeed we found the often reported interaction between evidence (absolute VD) and response accuracy where confidence responses ramp up and down for consistent and inconsistent decisions respectively (consistent: $\beta = 0.056 \pm 0.016$, $P < 0.001$, inconsistent: $\beta = -0.056 \pm 0.027$, $P = 0.017$, interaction evidence*consistency $\beta = 1.11 \pm 0.2$, $P < 0.001$; Fig. 1e, left). Moreover, we also found that confidence was higher for faster RTs ($\beta = -0.37 \pm 0.013$, $P < 0.001$; Fig. 1e, center). In line with the quantitative results in Fig. 1b, a qualitative inspection of the data shows that TV had a positive influence on confidence reports (Fig. 1e, right), which is not a direct prediction of the canonical model of confidence given its relative lack of weight on choice consistency, but it has previously been observed in value-based decision studies[4,32]. Thus, while most of the canonical signatures of confidence are qualitatively reflected in the data, our initial set of quantitative analyses suggest that confidence is influenced by factors other than uncertainty in the estimated input values for decision making.

**Generative dynamical models of confidence.** In order to gain a mechanistic understanding underlying these descriptive results, we implemented a modeling approach allowing us to tease apart the different components of the decision process and how these are related to confidence reports. It has been shown that subjective value reports can be directly used as input evidence in dynamic accumulation models, which can successfully explain both choices and RTs[28,33,39,40]. Moreover, related work in the perceptual domain suggests that information of both RTs and evidence strength provide relevant information about decision confidence[5,10]. In addition, given that during value-based choices decision-makers foveate—often repeatedly via changes in eye-fixation—to one of the choice options at a time, the degree of attentional effort during the decision process (formally defined below) has a significant impact on decision-making processes[28,36,41].

In order to jointly account for these factors, here we make use of a recent evidence accumulation model that takes into account attentional effort and does an excellent job at explaining value-based decisions while jointly considering RTs and fixation patterns on a trial by trial basis: the Gaze-weighted Linear Accumulator Model (GLAM)[41]. In this model, the effects of attention are captured by parameter $\theta$, which can be interpreted as a cognitive process controlling the degree of effort the agent

exerts to keep the evidence for the non-fixated choice option in working memory (henceforth, attentional effort). If $\theta = 0$, the agent completely ignores the evidence for the non-fixated choice item. But if $\theta = 1$ the agent exerts maximum effort and the two choice alternatives are equally weighted. The resulting gaze-weighted decision signals are then fed into a linear stochastic race (see Methods for details). In typical applications, the attentional effort $\theta$ is assumed to be agent-specific and a constant parameter in a given experiment[28,41]. Instead, here we considered the possibility that attentional effort fluctuates from trial to trial and studied the potential impact of such fluctuations on confidence reports. To this end, we considered two families of generative models of confidence: (i) a heuristic model, and (ii) a normative model.

The heuristic confidence model is based on the classic "balance of evidence" approach[42]. In brief, this approach proposes that the observer is conceived as accumulating successive differences between momentarily registered noisy values of two alternatives $i$ and $j$. As soon as one of the total accumulated differences reaches a decision bound $B$, the observer makes a decision in favor of the winner alternative. A workable definition of confidence is based on the "balance of evidence", that is, based on the difference between the total evidence of the two accumulators $e_i(t)$ and $e_j(t)$ at decision time $t$. The closer the particle of the loser alternative is to the decision bound, the lower the confidence of the observer[32,39]. In this case, on any given trial confidence is simply defined as

$$\text{confidence} = B - e_i(t), \tag{1}$$

where $B$ is the bound of the accumulator and $e_i(t)$ is the location of the loser accumulator at decision time $t$ (Fig. 2a; see Methods for a more detailed explanation of the model). Interestingly, this heuristic implicitly considers the influence of RT, given that longer decision times imply a higher probability that the loser accumulator is closer to the bound (Fig. 2a, b). While readouts of confidence in this model are simple, it does not directly consider the statistics of the environment and the decision process hence it is labeled here as a heuristic model.

The normative confidence model is based on the statistical definition of confidence[5,11,16]. In the case of our model specification, it is assumed that the decision-maker estimates the probability that the decision is correct by using the information of the decision time and the location of the losing accumulator, alongside the parameters of the decision process and the contextual statistics of the task (see Methods). Given that the experimenter has no access to the exact evidence of the losing accumulator in each trial, the best estimation the experimenter can assume is the expected evidence. In this case, one must gather information about the statistics of the decision process, and marginalize out the variables of no interest (see Fig. 2c; Methods). In this case, confidence is defined as

$$\hat{p}(\text{correct}\,|\,C, t, \Omega) = \int_E \hat{p}(\text{correct}\,|\,\overrightarrow{e}, C, t, \Omega)\, p(\overrightarrow{e}\,|\,C, t, \Omega)\, de, \tag{2}$$

where $\Omega$ are the set of parameters of the GLAM (including attentional effort $\theta$), $\overrightarrow{e}$ are the evidence levels of the correct and incorrect decision alternatives at the decision time $t$ (thus $\overrightarrow{e}$ implicitly contains information of the bound $B$), and $C$ is the observed type of choice from the experimenter's perspective (correct or incorrect). The contextual association between the probability of being correct and the dynamics of the choice process is represented by $\hat{p}(\text{correct}\,|\,\overrightarrow{e}, C, t, \Omega)$ (see Eq. (12) in Methods).

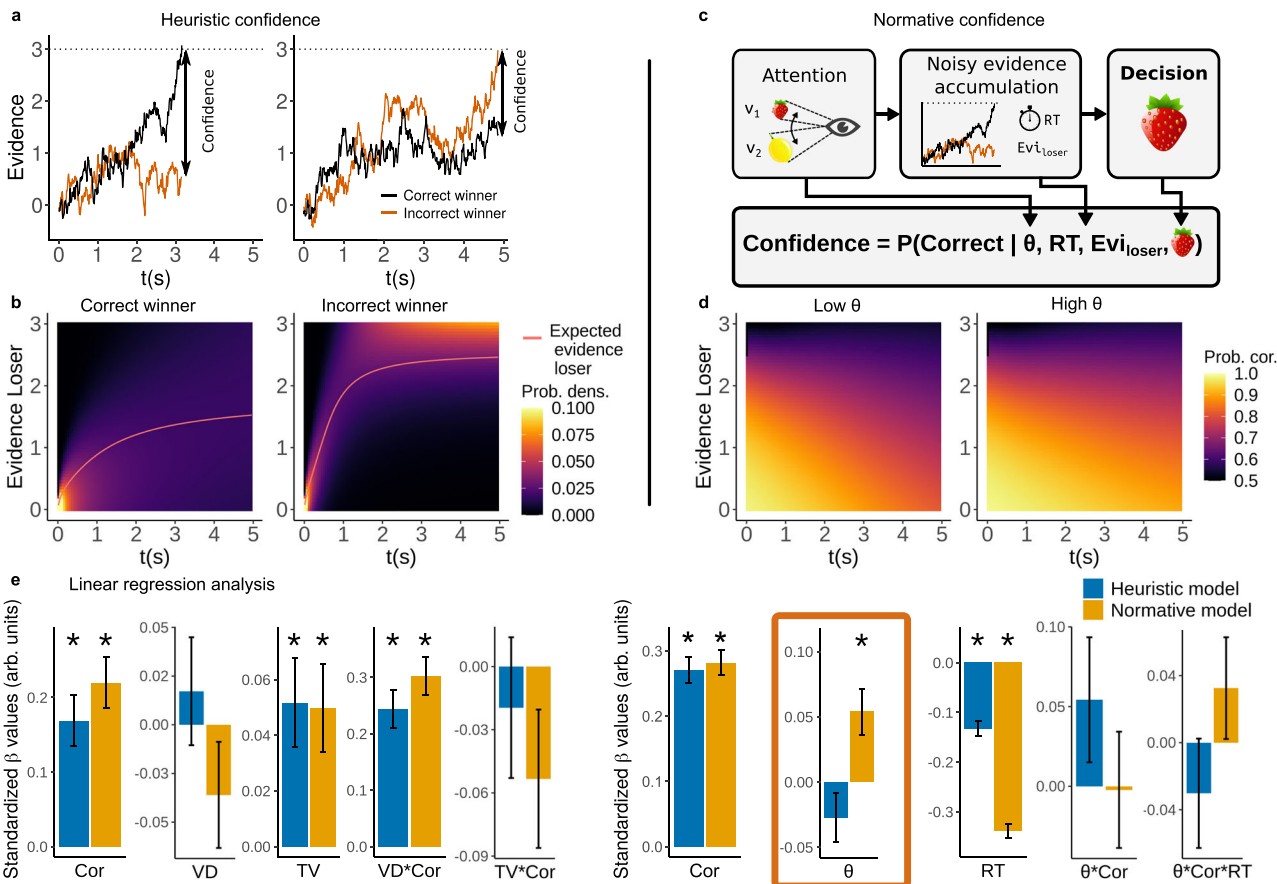

**Fig. 2 Generative modeling of confidence: heuristic and normative. a** Illustration of how confidence reports are generated by the decision-maker according to the heuristic process: confidence is simply computed as the difference between the decision bound and the evidence of the losing accumulator at the time of decision. **b** Confidence can be computed via the estimation of the expected evidence of the losing accumulator. **c** How the observer generates confidence reports according to the normative model: confidence is generated by computing the probability that the decision is correct given the decision time and the process model parameters. **d** Confidence predictions generated by the normative model as a function of RTs, evidence of the loser accumulator, and attentional effort. **e** Linear regression analysis of confidence comparing the heuristic (blue) versus the normative (orange) model. The results originate from two separate linear regressions, on the left Confidence ~ Correct (Cor) + VD + TV + VD*Cor + TV*Cor and on the right Confidence ~ Cor + $\theta$ + RT + $\theta$*Cor + $\theta$*Cor*RT. We use two linear regressions to prevent problems with high correlations between explanatory variables and to separate the input variables from the variables that are generated by the decision-maker. Bars indicate the mean of the standardized $\beta$ values and error bars the standard error, stars indicate significant difference from 0 with $\alpha = 0.05$. Statistics are calculated using $n = 33$ independent participants. Only the normative model predicts that confidence should be higher for higher values of attentional effort.

A contribution of our work is that based on our model specification, it is possible to generate predictions about how attentional effort $\theta$ during the decision process influences confidence reports. The two models (heuristic and normative) generate qualitatively similar predictions (Fig. 2e and Supplementary Figs. 4–6), including the interesting result that the higher the TV of the input alternatives, the higher confidence is, crucially after controlling for other relevant factors (Fig. 2e). This prediction is in line with our data (Fig. 1b, e) and recent work[4,39]. This result emerges in the GLAM model based on the fact that under the assumption that attentional effort is not always maximal, higher input values amplify the relative difference of the choice alternatives, furthermore higher input values result in faster RTs and therefore higher confidence. Interestingly, we find that only the normative model (and not the heuristic model) predicts that higher attentional effort $\theta$ leads to higher confidence reports (Fig. 2d, e). This prediction (alongside other slight qualitative differences between the two models, Fig. 2e and Supplementary Figs. 4–6) allows us to formally test whether confidence reports in our experiments are more favored by the heuristic model[32,39] or are better explained by the statistical definition of confidence[5,16].

**Trial-to-trial fluctuations in attentional effort influence confidence.** Based on these model predictions, we hypothesized that if participants use (or approximate) the statistical definition of confidence, then they should not only consider information about the strength of evidence and RT, but also attentional effort on a trial-to-trial basis. That is, we studied the possibility that human participants can introspect about how balanced (rational)—with respect to both items—they were during the comparison process, where high attentional effort (i.e., $\theta \approx 1$) should be related to high confidence. Crucially, we also considered in both heuristic and normative model specifications how fluctuations in the evidence gain $k$ (also known as drift rate, which is input value independent; see methods) may affect confidence as it has been described in the previous studies[32].

In order to evaluate the predictive power of the above-mentioned families of generative models and take potential trial-to-trial fluctuations in the relevant decision model parameters into account (both $\theta$ and $k$), we adopt the "joint modeling" approach[43] (Fig. 3a). This approach allows us to enforce reciprocal statistical relationships between the confidence reports and the parameters of the sequential sampling model by modeling

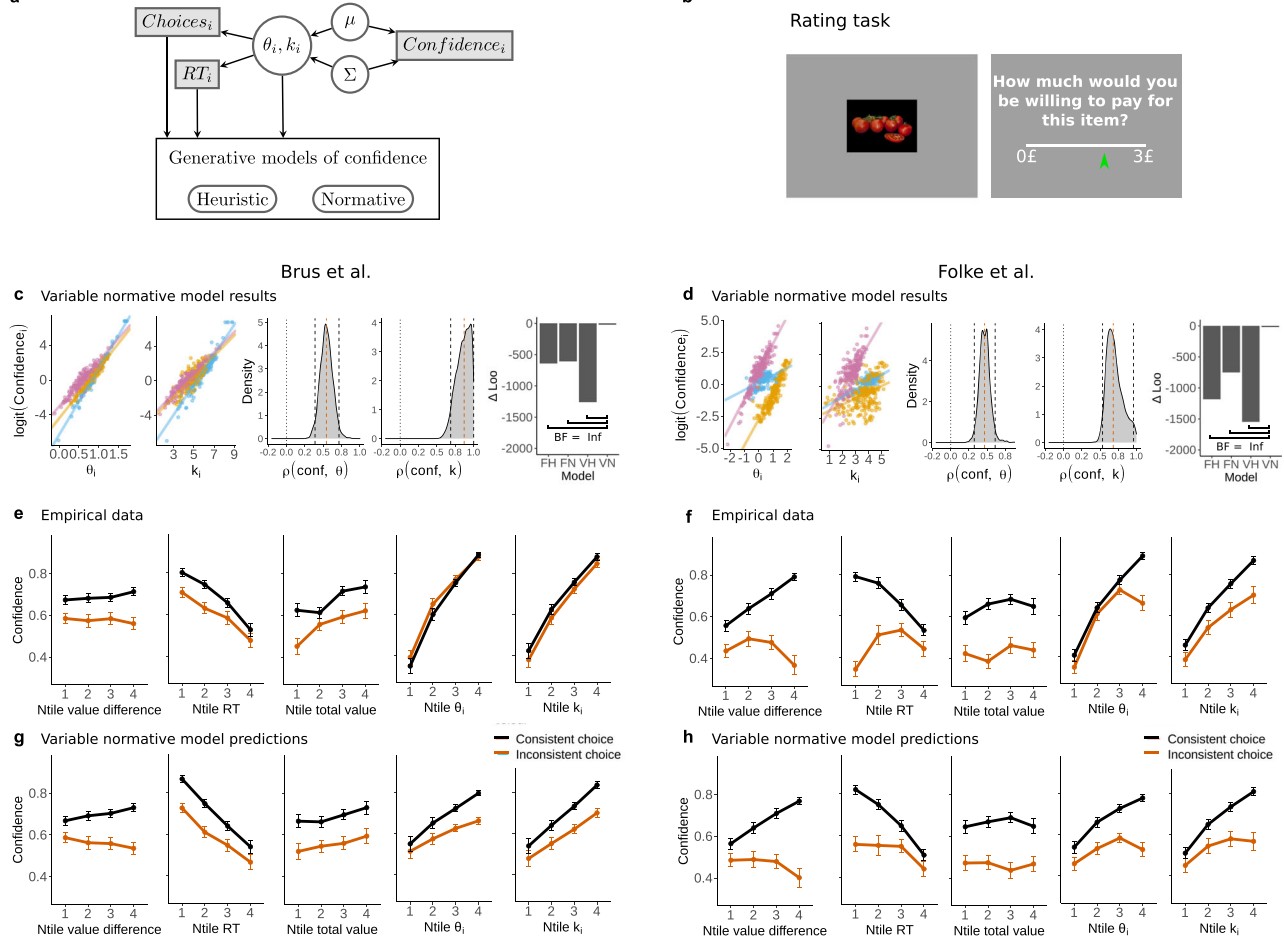

**Fig. 3 Joint modeling, the covariance approach. a** Graphical diagram for the joint model with the covariance approach. White circular nodes represent latent variables, grey rectangular nodes represent observable variables. The variables $\theta_i$, $k_i$, and choices and reaction times feed into the generative models of confidence. **b** Folke et al. performed similar experiments, with the key difference that the rating task is not repeated. Furthermore, subjects were asked how much they were willing to pay for a certain food item using a standard incentive-compatible Becker–DeGroot–Marschak method. **c** Results from the variable normative model and model comparison. Confidence is positively related to trial-to-trial fluctuations of attentional effort and evidence gain, shown for three example subjects (for all subjects see Supplementary Figs. 7 and 8). The estimated density of correlation parameters for confidence and attentional effort and for confidence and the evidence gain is bigger than zero. In both cases $\rho_{mcmc} > 0$ with $P < 0.001$. The vertical red dashed line indicates the median, black lines indicate the 95% confidence interval. Loo model comparison of the fixed heuristic (FH), variable heuristic (VH), fixed normative (FN), and variable normative (VN) model versions show that the VN model explains the data best. The Bayes Factor (BF) is calculated between the variable normative model and all other models, for all comparisons we find an infinite BF in favor of the variable normative model. **d** The same as (**c**), but for the Folke data. **e** The empirically found confidence levels as a function of (from left to right) value difference, reaction time, total value, attentional effort, and the evidence gain, split for consistent and inconsistent choice. Data are presented as mean values ± SEM. **g** The same as in e, but for the predictions of the variable normative model. **f, h** The same as in (**d**) and (**e**), but for the data of Folke et al. For the Brus et al. dataset $n = 33$ independent participants, for Folke et al. $n = 28$ independent participants. Source data are provided as a Source Data file.

these random variables simultaneously (Methods). Crucially, this approach also can automatically evaluate the statistical dependency between decision model parameters and confidence through correlation parameters (Methods). In addition, one can obtain approximate trial-to-trial readouts of the decision model parameters[43]. Here, it is important to highlight that this method is completely agnostic as to which mechanism generates the confidence reports (this method allows evaluating potential co-fluctuations between confidence and latent variables of the decision model). Therefore, this approach allows us to use information about the trial-to-trial readouts in the decision model and plug these values into the heuristic and normative definitions of confidence (Eqs. (1) and (2), respectively), thus allowing to obtain predictions of confidence and formally compare the predictive power of the two models (Fig. 3a).

First, we investigated which of the two models (heuristic or normative) can better predict future confidence reports assuming that attentional effort $\theta$ and evidence gain $k$ have fixed values (i.e., no trial-to-trial fluctuations in the values of these parameters as it is usually assumed in the literature[28,41]). While both models were monotonically related to the confidence reports ($\beta_{heuristic} = 0.45 \pm 0.04$, $P < 0.001$; $\beta_{norm} = 0.53 \pm 0.06$, $P < 0.001$), we found overwhelming evidence that the normative model provided more reliable predictions of confidence via model comparison (BF ≫ 1000 → ∞; also confirmed via leave-one-out (LOO) cross-validation metrics: ΔLOO = 29; Fig. 3c).

We then investigated whether models incorporating trial-to-trial-fluctuations according to the "joint modeling" approach provided more accurate predictions of confidence. We found significant trial-to-trial co-fluctuations between confidence and

attentional effort $\theta$ ($\rho = 0.54 \pm 0.08$, $P < 0.001$; Fig. 3c) and also between confidence and evidence gain $k$ ($\rho = 0.87 \pm 0.10$, $P < 0.001$; Fig. 3c). Subsequently, we used inferred trial-to-trial estimates of $\theta$ and $k$ to generate confidence predictions (Fig. 3a, Methods). Once again, we found that both models generate confidence predictions that are monotonically and significantly related to the empirical reports ($\beta_{\text{heuristic}} = 0.13 \pm 0.04$, $P < 0.001$; $\beta_{\text{norm}} = 0.61 \pm 0.05$, $P < 0.001$). But crucially, we found over-whelming evidence that the normative model provided more reliable predictions of confidence via model comparison (BF $\gg 1000 \to \infty$; also confirmed via LOO cross-validation metrics: $\Delta$LOO = 1260; Fig. 3c). Moreover, across the normative models, we found that the confidence model incorporating trial-to-trial fluctuations in attentional effort $\theta$ and evidence gain $k$ provide more reliable estimates of confidence reports than the models that assume fixed parameters across trials (BF $\gg 1000 \to \infty$ favoring the normative model with trial-to-trial variability in $\theta$ and $k$ across all model comparisons, Fig. 3c). Going beyond these quantitative analyses, importantly, we found that the qualitative predictions of the normative model closely match the empirical results (Fig. 3e, g; for completeness, we provide model predictions of the remaining models in Supplementary Fig. 4).

To rule out that the results obtained are due to the specifics of our design or cultural differences, we reanalyzed a value-based decision-making dataset from previous work (Folke et al.[4], Fig. 3b). Their experimental design is similar to our value-based task, except that participants performed only one round of ratings, and they were asked to indicate their willingness to pay for a given item; while in our task participants were asked to indicate how much they wanted to consume the item at the end of the experiment. Analyzing the data of this independent dataset, we fully replicate all the key results of our data. Using the models with fixed $\theta$ and $k$ parameters, we found that both models are monotonically related to the confidence reports ($\beta_{\text{heuristic}} = 0.51 \pm 0.06$, $P < 0.001$; $\beta_{\text{norm}} = 0.82 \pm 0.15$, $P < 0.001$), and found overwhelming evidence that the normative model provided more reliable predictions of confidence via model comparison (BF $\gg 1000 \to \infty$; also confirmed LOO cross-validation metrics: $\Delta$LOO = 333; Fig. 3d). Using the models allowing trial-to-trial fluctuations in $\theta$ and $k$, we found significant trial-to-trial co-fluctuations between confidence and attentional effort $\theta$ ($\rho = 0.47 \pm 0.08$, $P < 0.001$; Fig. 3d) and also between confidence and gain parameter $k$ ($\rho = 0.70 \pm 0.11$, $P < 0.001$, see Fig. 3d). Subsequently, we used inferred trial-to-trial estimates of $\theta$ and $k$ to generate confidence predictions. We found overwhelming evidence that the normative model provided more reliable predictions of confidence via model comparison (BF $\gg 1000 \to \infty$; also confirmed via LOO cross-validation metrics: $\Delta$LOO = 1548; Fig. 3d). Moreover, across the normative models, we found that the confidence model incorporating trial-to-trial fluctuations in atten-tional effort $\theta$ and evidence gain $k$ provide more reliable estimates of confidence reports than the models that assume fixed parameters across trials (BF $\gg 1000 \to \infty$ favoring the normative model with trial-to-trial variability in $\theta$ and $k$ across all model comparisons, Fig. 3d). Once again, we found that the qualitative predictions of the normative model closely match the empirical results (Fig. 3f, h; for completeness, we provide model predictions of the remaining models in Supplementary Fig. 5).

Taken together, these results strongly suggest that attentional effort is not a fixed model parameter but fluctuates from trial to trial (alongside the evidence gain) and human participants incorporate this knowledge during their confidence reports as predicted by the normative definition of confidence.

**Using confidence to infer potential latent variable fluctuations in static models.** The results presented above show how it is possible to capture trial-to-trial fluctuations in relevant latent variables of a given sequential sampling model (the GLAM in our case) based on the "joint modeling" approach, where both latent variables and confidence reports are treated as random variables (and not observed point estimates). However, this approach relies on the estimation of covariance matrices that are computationally demanding (in particular for models with hierarchical Bayesian structures). Moreover, in some cases, access to reaction time data and specification of dynamical models is not always possible in some datasets and studies of psychology and economics (never-theless, we emphasize that whenever possible, RT and decision data should jointly be used as they provide complementary information underlying decision processes[44]). Therefore, it might be of interest to investigate whether confidence reports—used in this case as observed (independent) variables—serve to infer potential fluctuations in latent variables in static decision-making models. We emphasize that in this case, it is not possible to establish generative models of confidence, but the goal is to study how observed confidence allows us to make inferences about the decision-making process.

We start by implementing a simple random utility model (RUM), which is a model that is widely used in the economics literature, that was extended to incorporate attentional factors[45] (Fig. 4a and Methods). Crucially, this extension allows for a straightforward interpretation of parameter estimates with only a fraction of the computational costs of commonplace attentional sequential sampling models[28,41] and the "joint modeling" approach. Therefore, this allows us to flexibly compare a range of model alternatives concerning factors such as attentional effort and other sources of noise in more complex models as we elaborate further below.

First, we studied whether attentional effort influences choices in the RUM, as demonstrated in previous work using sequential sampling models[28]. We find that participants discount the non-attended item ($\theta = 0.68 \pm 0.04$; Fig. 4b), thus reproducing previous reports and validating our model specification. Then, we investigated whether post-decision confidence reports predict trial-to-trial fluctuations of attentional effort $\theta$ and evidence gain $k$. We find that trial-to-trial confidence reports are related to the degree of attentional effort ($\beta^\theta = 0.80 \pm 0.06$, $P < 0.001$; Fig. 4f), and notably, attentional effort $\theta$ ranges over its full range from low to high confidence (Fig. 4b). In the same model, the trial-to-trial confidence is positively related to changes in evidence gain ($\beta^k = 11.3 \pm 3.8$, $P = 0.002$; Fig. 4d, f). The significant impact of confidence reports on these latent variables was confirmed by effect sizes significantly larger than zero (effect size $\beta^\theta = 13.3 \pm 2.0$, $P < 0.001$, effect size $\beta^k = 2.95 \pm 0.28$, $P < 0.001$; Fig. 4f). Qualitatively the model also captures the data well (Fig. 4h). In addition, cross-validation metrics revealed that the model that incorporates trial-to-trial fluctuations of $\theta$ and $k$ as a function of confidence explains the data more parsimoniously compared to the standard RUM models (Supplementary Fig. 9). This set of results mirror the ones obtained using the "joint modeling" approach, thus validating the usefulness of using confidence reports in order to reverse-engineer potential trial-to-trial fluctuations in latent variables of static decision-making models. For completeness, we applied the same inference approach, but this time using an attentional drift-diffusion model variant based on the RUM in order to further validate our results. We found nearly identical qualitative and quantitative results to those obtained using the static models (Supplementary Fig. 10).

Once again, in order to rule out that the results obtained are due to the specifics of our design or cultural differences, we reanalyzed the value-based decision-making dataset from previous work (Folke et al.[4]). This independent dataset fully replicates our findings that confidence reports are related to fluctuations in attentional effort

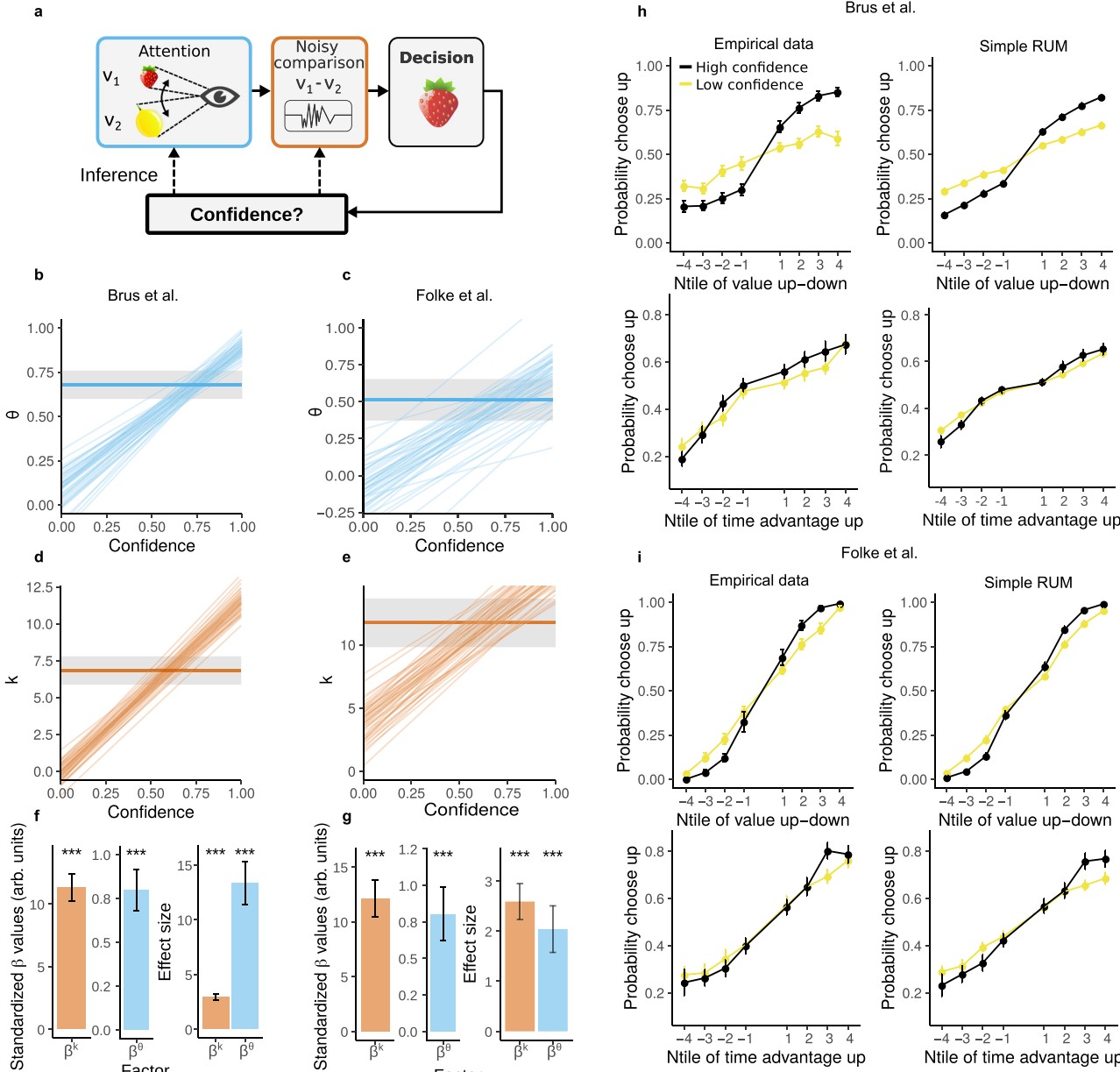

**Fig. 4 The RUM decision model. a** Sketch of the simple RUM decision model, color-coded to match the graphs. Observers infer the value of the food items by looking back and forth between choice alternatives. The subsequent comparison process is noisy. We investigate how confidence ratings influence trial-to-trial fluctuations of attentional factors and the evidence gain. **b, d** Comparison of parameter estimates of two alternative RUMs: a RUM with agent-specific estimates of $k$ and $\theta$ and a RUM that allows for trial-to-trial fluctuations of $k$ and $\theta$. b) the median of the posterior estimate of $\theta$ of the agent-specific RUM is indicated by the horizontal blue line, the shaded grey area indicates the 95% confidence interval. The diagonal blue lines represent 100 random samples of the posterior distribution of how $\theta$ changes with confidence in the RUM allowing for trial-to-trial fluctuations. Remarkably, $\theta$ changes over its full range as a function of confidence. d) the same as b, but for $k$. **f** Left: standardized posterior estimates of the relationship between confidence and $k$ and $\theta$. Error bars indicate the mean posterior estimate of the standard deviation. Both $\beta^k$ and $\beta^\theta$ are significantly bigger than zero with $P < 0.001$. Right: effect sizes of the results shown on the left. Error bars indicate the standard deviation of the posterior estimates of the mean of the effect size. Both the effect sizes of $\beta^k$ and $\beta^\theta$ are significantly bigger than zero with $P < 0.001$. P-values are based on the highest density interval of the posterior estimates. **h** Left column: the empirical probabilities of choosing the upper item; up: as a function of value difference; down: as a function of the difference in dwell time. Right column: the same as left but for the predicted probabilities of choosing the upper item by the simple RUM. The trials are median split in high/low confidence. Value difference and dwell time difference are split into eight groups of equal size. Data are presented as mean values ± SEM. **c, e, g, i**) Same as **b, d, f, h**, but for the data of Folke et al. For the Brus et al. dataset $n = 33$ independent participants, for Folke et al. $n = 28$ independent participants. Source data are provided as a Source Data file.

($\beta^\theta = 0.80 \pm 0.18$, $P < 0.001$, effect size $\beta^\theta = 2.04 \pm 0.47$, $P < 0.001$; Fig. 4c, g) and the evidence gain ($\beta^k = 12.1 \pm 1.7$, $P < 0.001$, effect size $\beta^k = 2.59 \pm 0.36$, $P < 0.001$; Fig. 4e, g), and can qualitatively capture the data well (Fig. 4i). Furthermore, cross-validation metrics revealed that the model that incorporates trial-to-trial fluctuations

of $\theta$ and $k$ as a function of confidence explains the data more parsimoniously compared to the standard RUM models (Supplementary Fig. 9). Thus, these converging results ensure that they are not a consequence of the specifics of our design or potential cultural differences.

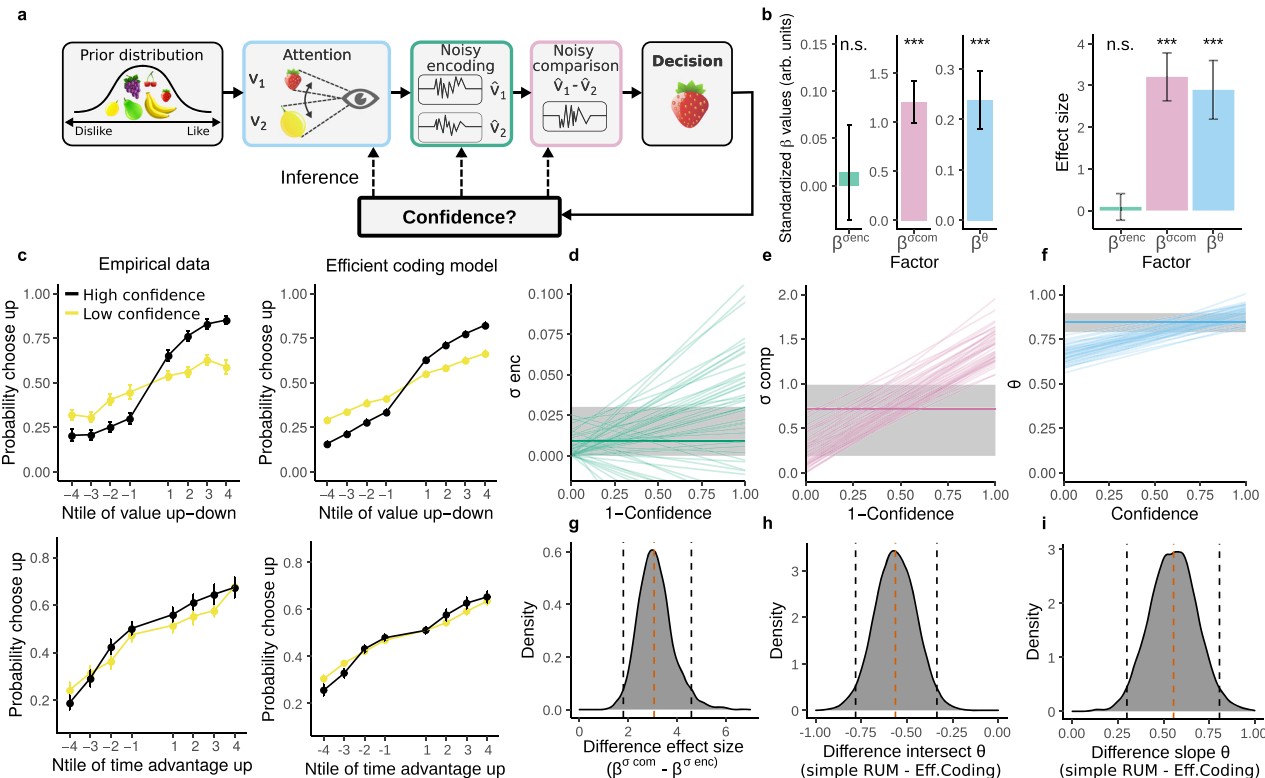

**Fig. 5 The efficient coding model. a** The decision process with three distinct process stages, color-coded to match the graphs. The prior matches the distribution of subjective values v of supermarket products. When choosing between two items, subjects look repeatedly at them, spending unequal time on the two options. The subjective values are internally encoded, the corresponding likelihood function $p(\hat{v}|v)$ is constrained by the prior $p(v)$ via efficient coding. Lastly, noise that occurs after the decoding is taken into account. **b** Standardized posterior estimates of the relationship between confidence and variance in the encoding process ($\beta^{\sigma_{enc}}$), the variance in the comparison process ($\beta^{\sigma_{comp}}$), and attentional factors ($\beta^{\theta}$). ($\beta^{\sigma_{enc}}$) is not significantly different from 0 ($P = 0.39$), both ($\beta^{\sigma_{comp}}$) and ($\beta^{\theta}$) are significantly bigger than zero with $P < 0.001$. The effect size of ($\beta^{\sigma_{enc}}$) is not significantly different from 0. Both the effect sizes of ($\beta^{\sigma_{comp}}$) and ($\beta^{\theta}$) are significantly bigger than zero with $P < 0.001$. Error bars indicate the mean posterior estimate of the standard deviation. *P*-values are based on the highest density interval of the posterior estimates. **c** Left column: the empirical probabilities of choosing the upper item; up: as a function of value difference; down: as a function of the difference in dwell time. Right column: the same as left but for the predicted probabilities by the efficient coding model. The trials are median split in high/low confidence. Value difference and dwell time difference are split into eight groups of equal size. Data are presented as mean values ± SEM. Source data are provided as a Source Data file. **d** Comparison of parameter estimates of two alternative efficient coding models: a model with agent-specific estimates of $\sigma_{enc}$ and a model that allows for trial-to-trial fluctuations of $\sigma_{enc}$. The median of the posterior estimate of $\sigma_{enc}$ of the agent-specific model is indicated by the horizontal green line, the shaded grey area indicates the 95% confidence interval. The diagonal green lines represent 100 random samples of the posterior distribution of how $\sigma_{enc}$ changes with confidence in the model allowing for trial-to-trial fluctuations. **e, f** Same as (**e**) but for $\sigma_{comp}$ and $\theta$. **g** Comparison of the effect sizes of the posterior estimates of $\sigma_{enc}$ and $\sigma_{comp}$. Vertical red dashed line indicates the median, black lines indicate the 95% confidence interval. **h** Comparison of the posterior estimates of the intercept of $\theta$ in the efficient coding model and the RUM. **i** Comparison of the posterior estimates of the slope of $\theta$ in the efficient coding model and the RUM. For the whole figure $n = 33$ independent participants.

**Confidence reports are not related to trial-to-trial fluctuations in reward value encoding.** The models presented above depart from the strong assumption that the experimenter has complete knowledge of the input stimulus value v for each choice alternative, where v is taken from the rating task and directly fed into decision models[4,28,32–35]. This strategy is warranted in studies of perceptual decision-making where the experimenter has full knowledge of the input values. However, this assumption is not ideal in studies of subjective value-based decisions, given that in this case, the experimenter has only limited access to the "true" input values v used to guide decisions. We account for this caveat by departing from the assumption that the observer needs to derive an estimate $\hat{v}$ of the items' value v to make the choice[26,46,47]. Note that this assumption acknowledges the common belief of the brain acting as an inference machine[48], which is not strictly the case in standard specifications of value-based decision models. Thus, we extend the RUM to model valuation as a probabilistic inference process

incorporating both encoding and decoding operations (efficient coding model, methods). The advantage of using this approach is that it allows taking into consideration: first, the statistics of the environment (i.e., the prior distribution) of subjective values for each individual; second, noisy encoding due to the limited capacity of systems to process information[26,27]; third, allows a parsimonious dissociation of noise at the encoding stage from downstream noise in the comparison process; and fourth, allows straight forward incorporation of information about fixation patterns (i.e., attentional effort). The computational extensions we introduce permit studying how post-decision confidence reports are related to each of these factors independently within one unified framework (Fig. 5a). Here, it is important to clarify that this model cannot be fitted to experiments that do not include information about rating variability (Methods). Therefore, this model cannot be fitted to the data from Folke et al., and in this part of the results, we restrict our analyses to our experimental data.

First, we investigated whether the efficient coding model also reveals attentional discounting independent of confidence. In this model, we found that participants also discount the non-attended item ($\theta = 0.85 \pm 0.03$; Fig. 5f), however to a lesser degree relative to the RUM (this result is discussed in further detail below). Also, this model can qualitatively capture the data well (Fig. 5c). Then, we used the efficient coding model to test the potential relationship between post-decision confidence reports and trial-to-trial fluctuations in encoding noise ($\sigma_{enc}$, which is input value specific), down-stream noise in the post-decoding comparison process ($\sigma_{comp}$, which input value independent), and attentional discount ($\theta$) (see Fig. 5a).

In line with the "model-free" analyses results (Fig. 1b), we found that confidence reports appear to be unrelated to trial-to-trial fluctuations in the encoding precision of the choice inputs ($\beta^{\sigma_{enc}} = 0.015 \pm 0.068$, $P = 0.41$; Fig. 5b, d). However, we found that confidence reports are strongly related to trial-to-trial fluctuations in down-stream comparison noise ($\beta^{\sigma_{comp}} = 1.21 \pm 0.22$, $P < 0.001$; Fig. 5e). The dissociation in the relationship between confidence reports and these two distinct noise sources was confirmed by a significant difference in their effect sizes ($\Delta\beta^{\sigma}_{EffectSize} = 3.18 \pm 0.71$, $P < 0.001$; Fig. 5g). These results are in line with the notion that integration noise of the input values for each choice alternative is not related to confidence reports during human value-based decisions as in our initial analyses (Fig. 1b) and is now confirmed via a normative inference model (Fig. 5).

In addition, we found that in this model confidence reports are also strongly related to the degree of attentional effort ($\beta^{\theta} = 0.24 \pm 0.06$, $P < 0.001$; Fig. 5b). Interestingly, confidence appears to induce a smaller (but still highly significant) influence on $\theta$ in the efficient coding model relative to the RUM. This is evident by the fact that $\beta^{\theta}$ was steeper in the RUM than in the efficient coding model ($\Delta\beta^{\theta} = 0.55 \pm 0.13$, $P < 0.001$; Fig. 5i) and additionally, the intercept parameter $\theta_{base}$ (i.e., the value of $\theta$ for the lowest level of confidence, see Methods) has a higher value in the efficient coding model than in the RUM ($\Delta\beta^{\theta}_{base} = -0.56 \pm 0.11$, $P < 0.001$; Fig. 5h). This result reveals that incorporating input-specific uncertainty in the decision model reduces the influence of attentional effort $\theta$. It follows that participants are more "rational" than observed with models where this input-specific uncertainty is not considered. Therefore, our formal and more statistically complete inference model highlights the importance of incorporating input-specific uncertainty in process models of preference-based behavior—a fundamental aspect classically considered in perceptual inference processes[38,49].

## Discussion

We implemented a behavioral task and a computational modeling approach that allowed us to dissect and understand how distinct components of the value-based choice process and associated sources of noise related to choice behavior and post-decision confidence reports. Following the statistical computation definition that confidence should reflect choice consistency, one would expect that all factors in the decision process that affect decision variability and certainty should have a corresponding impact on confidence reports[11]. For instance, we find that TV has no effect on choice consistency but does affect confidence ratings. While initially surprising, this result appears in line with observations from perceptual decision-making studies showing that sources of evidence supporting the correct stimulus identification response (i.e., positive evidence) and sources of evidence that support the alternative response (negative evidence) both influence choice accuracy, but positive evidence affects confidence more strongly[19,20]. However, based on the experimental procedure

adopted in our study, we cannot ascertain whether the true factor underlying this phenomenon is TV or positive evidence as by design these two factors are correlated (see Supplementary Fig. 11).

A second surprising finding is that encoding precision of reward value inputs has an impact on choice (as it is expected from canonical inference models), but no impact on confidence reports. This result was initially reflected in our model-free analyses, revealing that rating variability negatively impacts choice consistency but not confidence. However, this analysis did not allow us to directly determine whether trial-to-trial fluctuations in encoding precision were related to post-decision confidence reports. Therefore, we implemented an inference model allowing us to, not only, formally test the role of value encoding precision of the input options, but also to separate encoding noise from downstream noise in the comparison process[26]. This model confirmed that trial-to-trial fluctuations in the encoding precision of reward values are not related to post-decision confidence reports. However, we found that sources of noise independent of input values substantially impact confidence reports. These results reveal an important aspect of how confidence emerges in economic decisions, which is not fully congruent with the normative view that confidence reflects an optimal Bayesian estimate that the decision is correct[11,12] nor with the idea that confidence is a direct transformation of evidence strength[13–15]. Nevertheless, we acknowledge that there might be other processes that our model does not directly capture, but may also influence the variability of confidence reports, such as how confidence can dynamically change during choice (for instance due to dynamic adjustments of decision bounds[50]), and also after a decision is made[10,51].

Regarding our established dissociation between encoding and comparison noise, at first sight, this result seems at odds with recently reported results by Castanon et al.[52]. It was found that humans disregard downstream integration noise while using encoding noise to form believes about confidence[52]. Here, we argue that the apparent difference between the influence of encoding noise on confidence in our study and the study of Castanon et al. has its basis in the definition of encoding noise and the characteristics of the task. Castanon et al. use a categorization task where they present multiple tilted gratings in a circular array and vary either the contrast level of the gratings (resulting in different levels of encoding noise) or the variability of the gratings' orientations (resulting in different levels of integration noise, which they define as late noise). In their task subjects are asked to categorize the average orientation as tilted clockwise or counterclockwise. First, we argue that what the authors of this work define as integration noise, could be very much related to what we define as encoding noise in our reward task. It has been suggested and is commonly accepted that reward value is formed via the integration of different discrete components (e.g., memories and emotions) associated to the physical features of the choice alternatives[25]. Therefore, it is well possible that trial-to-trial fluctuations in reward value estimations and their degree of variation is related to the definitions of integration noise in the above-mentioned categorization tasks. This would be in principle congruent with a previous study in the perceptual decision-making domain, demonstrating that observers underestimate the variance of orientation noise, which leads to distorted confidence reports[8]. Second, in our task, observers integrate information of two distinct choice alternatives thus resulting in two distinct value estimates that need to be compared via downstream circuits. However, typically in categorization tasks observers integrate multiple cues and generate a single value estimate to be categorized. Therefore, we argue that perceptual decision tasks of the kind discussed above would resemble the

characteristics of our task only in the event where feature integration of two distinct feature estimates takes place simultaneously (e.g., the comparison of two distinct arrays of tilted gratings) and then compared according to an abstract decision rule (e.g., which of the mean estimates is closer to cardinal). It will be interesting to investigate in future work whether a perceptual task designed in this way would also reveal that downstream comparison noise heavily influences confidence reports.

A further contribution of our work is the finding that the degree of attentional effort in the comparison process is not just a static subject-specific characteristic as it is usually assumed in the literature[28,36,41], but it is a rather highly dynamic cognitive function that fluctuates considerably from trial-to-trial. Interestingly, we found that subjects can introspect about their attentional effort which is subsequently reflected in their confidence reports. In other words, participants can detect whether they exerted enough effort to compare the two alternatives in a balanced way (resulting in high confidence judgments), or whether in any given trial they were "lazy" (that is, lacked effort) and did not pay enough attention to the options available. This might be surprising in the light of some studies from the perceptual decision-making domain indicating that people overestimate their perceptual sensitivity for unattended stimuli and peripheral vision in general[53–56]. This might lead us to believe that metacognitive performance is poor when peripheral vision is concerned and one could hypothesize that since the quality of peripheral vision is overestimated, confidence ratings based on peripheral vision are overestimated too. We have not explicitly tested for systematic overconfidence due to overestimation of the visibility of the unattended item. However, our results indicate that metacognition does correctly include information about peripheral vision in the sense that participants track the amount of effort they spend to incorporate the unattended item into the decision process and that more effort correlates with higher confidence ratings.

We speculate that trial-to-trial variations in attentional effort are related—at least to some extent—to fluctuations of working memory utilization during the comparison process. Recall that when an observer fixates one option, she needs to uphold the memory of the non-fixated option, which requires higher effort, but also results in a more balanced comparison and eventually more rational choice. Thus, we argue that there must exist a tight relationship between our definition of attentional effort with working memory and confidence as has been suggested in previous work[57,58].

Our findings extend the intricate relationship between the accumulation of evidence for decisions and confidence reports[50,59]. However, our results additionally indicate that not only is confidence used as an online control process, setting bounds on evidence accumulation, nor is it just a reflection of the quality of the evidence. Confidence also reflects trial-to-trial fluctuations in the amount of effort exerted during the accumulation process. Our results also indicate that metacognitive processes appear to be blind to the irreducible uncertainty that emerges during the integration of valuable information, suggesting that people acknowledge their capacity limitations to process information[27,60] and rather use late-stage processes of the decision formation to guide their metacognitive processes. Indeed, this argument is generally in line with a range of studies suggesting that confidence depends on late-stage processing[1,6,51,61–65].

Taken together, our study shows compelling evidence that we can introspect about how much attention we pay to the choice options available during our decisions, thus, revealing a mechanistic interplay of endogenous attentional effort and reward values for guiding decisions and metacognitive awareness of choice certainty. We argue that our findings might not be limited to the domain of value-based decisions but might have important implications for refining models of metacognitive distortions in psychiatric disorders[66], where the interaction between attention, cognitive effort, and goal-directed processes play a key role in the characterization of diverse psychopathologies[67].

## Methods

**Participants**. The study tested healthy young volunteers ($n = 35$, age 19–37 years). However, due to bad eye-tracking recording quality, two subjects have been excluded from the analysis. The sample size was determined based on previous studies using similar stimuli and tasks[26,33,34]. Participants were instructed about all aspects of the experiment and gave written informed consent. None of the participants suffered from any neurological or psychological disorder or took medication that interfered with participation in our study. Participants received monetary compensation for their participation in the experiment (20 CHF/h), in addition to receiving one food item after the choice task (see below). The experiments conformed to the Declaration of Helsinki and the experimental protocol was approved by the Ethics Committee of the Canton of Zurich.

In addition to the data collected in our lab, we also analyze the choice data from Folke et al.[4]. We used the data from experiment 1 in their study in an attempt to replicate the contributions of our work. The dataset consists of $n = 28$ healthy young volunteers.

**Behavioral task**. Our experiment consisted of three main phases: (1) rating phase 1, (2) rating phase 2, and (3) the choice task. In rating phase 1, we asked the participants to provide subjective preference ratings for a set of 64 food items using an on-screen slider scale (Fig. 1a). All of the food items were in stock in our lab and participants were notified about this. Importantly, participants saw all food products before the ratings so that they could effectively use the full range of the rating scale. Moreover, participants knew that all products were randomly drawn from the two biggest supermarket chains in Switzerland. Based on previous studies in our lab[26,33,34], we selected food items that varied all the way from items that most participants would find unappealing (e.g., cucumber) to items that most participants would find highly appetitive (e.g., ice cream). This was important as our model should capture the full range of subjective values that humans typically assign to food items on a daily basis.

During the ratings, participants indicated "how much they want to eat the presented food item at the end of the experiment". Participants were informed that the rightmost endpoint would indicate items that they would most love to eat, whereas the leftmost endpoint would indicate items that they would most hate to eat. The initial location of the slider was randomized for each item to reduce anchoring effects.

Rating phase 2 was identical to rating phase 1 and took place immediately after phase 1. The order of the items' presentation was randomized. Crucially, participants were not informed before rating phase 1 that a second rating phase and a decision-making task would take place. This was important as it prevented participants from actively memorizing the location of the rating in the slider in the first phase, thus providing us with a clear measure of the variability in the value estimates.

Immediately after the two rating phases, a custom made algorithm selected a balanced set of decision trials divided into four VD levels on the rating scale (rating difference 5, 10, 15, and 20% of the length of the rating scale), as defined by the average rating across phases 1 and 2 provided by each participant. Decision-making trials started with a central presentation of a fixation cross for 1–2 s. Immediately after this, two food items were displayed simultaneously, one in the upper and one in the lower part of the screen (Fig. 1a). The food items were presented until response and participants had up to four seconds to make a choice. Participants were instructed to choose which of the two items (upper or lower) they preferred to consume at the end of the experiment. To make these choices, participants pressed one of two buttons on a standard keyboard with their right index finger (upper item) or their right thumb (lower item). We defined a consistent choice as a trial in which the subject chose the item with a higher mean rating from the prior rating phase. Each experimental session comprised a maximum of 240 trials (this depended on the rating distribution of each participant) divided into 6 runs of 40 trials each. The trials were fully balanced across rating-difference levels and location of consistent response options (Up or Down).

Participants' eye movements were recorded throughout the choice task at 1000 Hz with an EyeLink 1000 Plus eyetracker (SR Research). To make sure that participants deliberated between the two alternatives, we excluded trials where participants had not fixated on every option available at least once. Based on the poor quality of the eyetracking data during the whole session, two participants were excluded from the analyses, thus resulting in a final sample of $n = 33$ subjects.

After each choice, participants indicated their confidence in their decision on a continuous rating scale. We informed participants that the leftmost side of the confidence rating scale means "Not at all" confident and the rightmost side means "Totally" confident. Neither choices nor confidence ratings were time-constrained. The experiment was implemented in Matlab R2016b with the use of Psychtoolbox.

The dataset that we reanalyze here from Folke et al.[4] is similar to our value-based task, except for the following main aspects. First, in their rating task participant were instructed to indicate their willingness to pay for a given item; in our task, we asked them to indicate how much they wanted to consume the item. Second, in their experiment they collected one round of ratings; we collected two. Here it is important to mention that the lack of more than one rating does not allow us to test all effects that are specific to item-specific variability in value coding in the efficient coding model (see below). However, all other analyses related to the influences of attention can be studied in both tasks. Crucially, these differences between the two tasks allow us to rule out that the results obtained in our experiment are due to specific aspects of the design or cultural differences between Swiss and British food decisions.

**Generative models of confidence.** In this section, we describe (i) the sequential sampling model used to jointly account for RTs and choices, (ii) the generative models of confidence based on the sequential sampling model, and (iii) the "joint modeling" approach used to estimate trial-to-trial fluctuations in the relevant latent variables of the decision process which are then used as inputs to the generative models.

*Sequential sampling model.* In order to implement the generative models of confidence, we make use of the GLAM, which allows incorporating RTs, choices, and gaze information in the decision process[41]. In brief, the GLAM belongs to the class of race models and assumes noisy accumulation in favor of each alternative $i, j$, where choices are determined once the winner accumulator reaches a decision boundary $B$. We define the relative accumulated evidence in favor of alternative $i$ as

$$e_i(t) = e_i(t-1) + kR_i + \varepsilon, \qquad (3)$$

with $e_i(0) = 0$ and $\varepsilon \sim N(0, \sigma^2)$, where $\sigma$ is the standard deviation of an unbiased normally distributed noise, $k$ is the drift rate and $R_i$ is the average amount of relative evidence for alternative $i$ at each time point $t$.

We define the absolute evidence signal $S_i$ for alternative $i$

$$S_i = g_i v_i + (1 - g_i)\theta v_i, \qquad (4)$$

where $v_i$ is the subjective value readout based on the average ratings of each alternative $i$, $g_i$ is the proportion of time that the agent looks at item $i$ on each trial, and $\theta \leq 1$ is the attentional effort parameter, which determines the strength of down-weighting when the agent does not fixate item $i$. If $\theta = 1$, then the agent exerts high effort to keep item $i$ in memory when she is not looking at it. On the other hand, if $\theta = 0$, then the agent ignores the evidence of the non-fixated item at any given moment during the choice process. This suggests that for values of $\theta$ closer to 1 the agent is being fully rational in her choices, which additionally could be interpreted as exerting effort to keep the unattended items in working memory. On the other hand for values of $\theta$ closer to zero, the agent is being "lazy" and tends to ignore the unattended alternative in the comparison process.

The GLAM assumes an adaptive representation of the relative evidence signals via

$$s(x) = \frac{1}{1 + \exp(-\tau x)}, \qquad (5)$$

where $\tau$ is a scaling parameter and $x \equiv S_i - S_j$, and finally $R_i$ in Eq. (3) is defined as $R_i \equiv s(x)$.

The first passage time density of a single accumulator $e_i$ with decision boundary $B$ is given by

$$f_i(t) = \left[\frac{\lambda}{2\pi t^3}\right]^{\frac{1}{2}} \exp\left\{\frac{-\lambda(t-\mu)^2}{2\mu^2 t}\right\}, \qquad (6)$$

with

$$\mu \equiv \frac{B}{kR_i} \text{ and } \lambda \equiv \frac{B^2}{\sigma^2}. \qquad (7)$$

Therefore, the probability that accumulator $e_i$ crosses $B$ at time $t$ before accumulator $e_j$ is given by

$$p_i(t) = f_i(t)(1 - F_j(t)), \qquad (8)$$

where $F()$ is the cumulative distribution function of $f()$, which is given by

$$F(t) = \Phi\left(\sqrt{\frac{\lambda}{t}}\left(\frac{t}{\mu} - 1\right)\right) + \exp\left(\frac{2\lambda}{\mu}\right)\Phi\left(-\sqrt{\frac{\lambda}{t}}\left(\frac{t}{\mu} + 1\right)\right), \qquad (9)$$

where $\Phi()$ is the standard normal cumulative density function.

*Heuristic confidence model.* The heuristic confidence reports are based on the location of the loser accumulator in any given trial. The closer the particle of the loser alternative is to the decision bound, the lower the confidence of the observer[32,39]. In this case, on any given trial confidence is defined as (Eq. (1) in the main text)

$$\text{confidence} = B - e_i(t),$$

where $B$ is the bound of the accumulator and $e_i(t)$ is the location of the loser accumulator at decision time $t$ (or RT).

On any given trial the experimenter has information about the RT but has no access to the location of the loser accumulator, therefore we must marginalize over all possible locations of the loser accumulator. Based on the process model assumptions of the GLAM, it can be shown that the expected confidence for a particular RT is given by

$$E[\text{confidence} \,|e_i, t] = B - \left((kR_i)t + \frac{\phi(\beta)}{\Phi(\beta)}\sigma\sqrt{(t)}\right), \qquad (10)$$

with

$$\beta \equiv \frac{B - (kR_i)t}{\sigma\sqrt{t}}, \qquad (11)$$

where $\phi()$ is the density function of the normal distribution (see Fig. 2b).

*Normative confidence model.* The normative confidence model is based on the statistical definition of confidence. In this case, it is assumed that the decision-maker estimates the probability that the decision is correct by using the information of the decision time, alongside the parameters of the decision process and the contextual statistics of the task. More formally, one can calculate the log-posterior odds of a correct response for all possible combinations of RTs and decision variables utilized by the observer in a given context or environment

$$\log\left[\frac{p(a_1|\overrightarrow{e}, t)}{p(a_2|\overrightarrow{e}, t)}\right] = \log\left[\frac{\sum_i p(\overrightarrow{e}, t|a_1, \Omega_i)p(\Omega_i)}{\sum_i p(\overrightarrow{e}, t|a_2, \Omega_i)p(\Omega_i)}\right], \qquad (12)$$

where $\Omega$ are the set of parameters of the decision model, $\overrightarrow{e}$ are the evidence levels of the correct and incorrect decision alternatives at the decision time $t$, and $a_1$ and $a_2$ are the correct and incorrect decision alternatives, respectively (see also Fig. 2c).

Once again, given that the experimenter has no access to the exact evidence of the losing accumulator in each trial, the best estimation the experimenter can assume is the expected evidence. In this case, one must gather information about the statistics of the decision process, and marginalize the location of the loser. In this case, confidence is defined as (Eq. (2) in the main text)

$$\hat{p}(\text{correct} \,|C, t, \Omega) = \int_E \hat{p}(\text{correct} \,|\overrightarrow{e}, C, t, \Omega)p(\overrightarrow{e}|C, t, \Omega)\, de,$$

where $C$ is the observed type of choice from the experimenter's perspective (correct or incorrect) and $\hat{p}(\text{correct} \,|\overrightarrow{e}, C, t, \Omega)$ represents the contextual association between the probability of being correct and the dynamics of the choice process.

*Joint modeling approach.* In order to derive confidence predictions based on the generative models defined above, we implemented the "joint modeling" approach[43], which allows us to enforce reciprocal statistical relationships between the confidence reports and the parameters of the sequential sampling models by modeling these random variables simultaneously. Specifically, we adopt a "covariance approach" which allows describing the joint distribution of the decision model parameters $\Omega$ and the confidence reports $c$ through a statistical constraint, and crucially, where confidence is not treated as a fixed point estimate, but as a random variable. That is, we impose an overarching distribution governed by parameters $\Psi$ which are used to describe the patterns of the joint distribution $(\Omega, c)$. This is achieved via a linking function $M$ with parameters $\Psi$

$$(\Omega, c) \sim M(\Psi). \qquad (13)$$

Here, we assume that the linking function $M$ is given by the multivariate normal distribution, and the goal in this statistical model is to find the hyperparameters of the mean vector $\boldsymbol{\mu}$ and the variance-covariance matrix $\boldsymbol{\Sigma}$. For instance, if one would like to investigate potential trial-to-trial co-fluctuations between attentional effort $\theta$, evidence gain $k$, and confidence $c$, the goal is to find the set of hyperparameters

$$\boldsymbol{\mu} = \begin{pmatrix} \mu_c \\ \mu_\theta \\ \mu_k \end{pmatrix} \qquad (14)$$

$$\boldsymbol{\Sigma} = \begin{pmatrix} \sigma_c^2 & \rho_{c\theta}\sigma_c\sigma_\theta & \rho_{ck}\sigma_c\sigma_k \\ \rho_{c\theta}\sigma_c\sigma_\theta & \sigma_\theta^2 & \rho_{k\theta}\sigma_k\sigma_\theta \\ \rho_{ck}\sigma_c\sigma_k & \rho_{k\theta}\sigma_\theta\sigma_k & \sigma_k^2 \end{pmatrix} \qquad (15)$$

Conveniently, this method allows to automatically evaluate the statistical dependency between decision model parameters and confidence through the

correlation parameter $\rho$. It was implemented in a Bayesian framework using JAGS in R. The model has nine parameters that need to be estimated, we placed uninformative priors between sensible limits on all parameters as follows:

$$
\begin{aligned}
\mu_c &\sim Normal(0, 10^{-5}) \\
\mu_\theta &\sim Normal(0, 10^{-5}) \\
\mu_k &\sim Normal(0, 10^{-5}) \\
\sigma_c &\sim Uniform(0, 10) \\
\sigma_\theta &\sim Uniform(0, 10) \\
\sigma_k &\sim Uniform(0, 10) \\
\rho_{c\theta} &\sim Uniform(-1, 1) \\
\rho_{ck} &\sim Uniform(-1, 1) \\
\rho_{k\theta} &\sim Uniform(-1, 1)
\end{aligned}
\tag{16}
$$

*Confidence predictions and model comparison.* We obtained trial-to-trial predictions in parameters $\theta$ and $k$ by sampling from the distribution

$$
(c, \theta, k) \sim M(\boldsymbol{\mu}, \boldsymbol{\Sigma}).
\tag{17}
$$

Predictions of $\theta$ and $k$ were then plugged into the confidence generative models described above—alongside all other GLAM parameters fitted using the joint approach—in order to derive confidence predictions on a trial-by-trial level. In order to account for differences in the use of the confidence rating scale, the predicted values were re-scaled via a simple linear regression. The predictions from these models were used to generate the qualitative predictions presented in Fig. 3e, g and Supplementary Figs. 4–6. Model comparison was carried out via computation of the BF estimated based on the bridge sampling approach using the brms package in R[68]. For completeness, the LOO information criterion was also computed.

**Reverse inference of latent variables based on confidence.** We investigated whether confidence reports—used in this case as observed (independent) variables—serve to back-engineer potential fluctuations in latent variables of static decision-making models (i.e., models that do not explicitly consider RT information). We emphasize that in this case, it is not possible to establish generative models of confidence, but the goal is to study how observed confidence allows us to make inferences about the decision-making process.

*Random utility model (RUM).* In the standard RUM, the agent faces two options $i$ and $j$, where each option has a subjective value $v_i$ and $v_j$, respectively. In standard neuroeconomic experiments, $v$ is assumed to be the outcome of their willingness to pay for a given option[4], or the desirability to consume a given item[26,28]. In standard RUMs, it is usually assumed that a given option $i$ is corrupted by some general noise $\eta_i$, with the usual assumption $\eta_i \sim N(0, \sigma_i)$. If $\sigma$ is assumed to be constant for all universe of goods in a given context, then the probability of choosing option $i$ over $j$ is given by

$$
P(v_i > v_j) = \Phi\left(\frac{v_i - v_j}{\sigma\sqrt{2}}\right).
\tag{18}
$$

If one would like to take into account the fact that when making decisions, people tend to look back and forth between choice alternatives, then one must take into consideration these potential shifts in attention between the choice alternatives. Recently, Smith et al. developed a variant of the RUM that allows taking into account the proportion of time that participants spend looking at a pair of choice alternatives[45]. Let $A_i$ be the fraction of time spent looking at option $i$ (with $A_j = 1 - A_i$). Then it can be shown that the probability of choosing option $i$ over $j$ while considering the effects of attention is given by

$$
P(v_i > v_j) = \Phi\left(\frac{A_i v_i - A_j v_j + \theta(A_j v_i - A_i v_j)}{\sigma\sqrt{2}}\right),
\tag{19}
$$

where $\theta \leq 1$ is defined here as the attentional effort parameter.

Given that in our study we use a hierarchical Bayesian data analyses framework, this allows the convenient possibility of studying the effects of an observed variable (e.g., confidence) on a latent variable (e.g., $\theta$). For simplicity, we investigate such influences via linear relationships. Thus, in order to study the link between confidence reports and attentional effort, we assume

$$
\theta_n = \theta_{\text{base}} + \beta_s^\theta * c_n,
\tag{20}
$$

where $\beta_s^\theta$ is the effect of confidence $c_n$ on attention in a given trial $n$. The subscript $s$ denotes that the effect is participant-specific which is modeled as a random-effects factor under the assumption that it is drawn from a population distribution $\beta_s^\theta \sim N(\beta^\theta, \sigma^\theta)$, where $\beta^\theta$ and $\sigma^\theta$ determine the mean and the s.d. of the population distribution, respectively. In this case, positive values of $\beta^\theta$ would indicate that higher levels of attentional effort $\theta$ are reflected in higher levels of confidence.

Likewise, we can study the relationship between confidence and reward value variability as defined by the RUM ($\sigma$ in Eqs. (18) and (19)), which directly influences the consistency of the choices). Defining $k \equiv \frac{1}{\sigma\sqrt{2}}$, we study the influence of trial-to-

trial fluctuations of the evidence gain $k$ on confidence as follows

$$
k_n = k_{\text{base}} + \beta_s^k * c_n.
\tag{21}
$$

As before, the possible influence of confidence $\beta_s^k * c_n$ is drawn from a population distribution $N(\beta^k, \sigma^k)$. Here, we list the uninformative priors placed the population distribution

$$
\begin{aligned}
\theta_{\text{base}} &\sim Normal(0, 10^{-10}) \\
\sigma^{\theta_{\text{base}}} &\sim Uniform(10^{-10}, 5) \\
\beta^\theta &\sim Normal(0, 10^{-10}) \\
\sigma^\theta &\sim Uniform(10^{-10}, 5) \\
\beta^k &\sim Normal(0, 10^{-10}) \\
\sigma^k &\sim Uniform(10^{-10}, 5)
\end{aligned}
\tag{22}
$$

*Efficient RUM.* Different from the simple RUM, here we assume that the input reward values $v$ in the inference process are not the observed ratings, but rather the most likely input values of a resource-constrained generative model that leads to the generation of noisy value estimates $\hat{v}$ This is an important consideration that is different from classical approaches in perceptual decision making since experimenters have no direct access to the "true" value $v$ of the presented object to an observer. Here we assume that this "true" value $v$ has been shaped by each observer's personal history of experiences with this type of object and is therefore entirely subjective. Based on studies supporting the notion of contextual adaptation of valuation circuits given capacity limitations, we assume optimal use of the underlying neuronal scale to represent reward values given the expected/learned natural distribution of values in the given environment, i.e., the prior $p(v)$. Under assumptions of mutual information maximization at the encoding stage and minimization of the Bayesian mean squared error at the decoding stage, one can obtain approximate expressions for the expected value and variance that explain the generation of noisy estimations $\hat{v}$ conditional on a particular input stimulus with value $v_0$

$$
E[\hat{v}|v_0] \approx v_0 + \phi'' \cdot \sigma_{\text{enc}}^2
\tag{23}
$$

$$
Var[\hat{v}|v_0] \approx (\phi')^2 \cdot \sigma_{\text{enc}}^2,
\tag{24}
$$

where $\sigma_{\text{enc}}$ is the noise of the limited system that encodes reward values (which we assume to be constant across the dynamic neural range, i.e., one free parameter), and $\phi$ is the quantile function of the prior $p(v; \omega)$, where $\omega$ are the parameters of the prior (for details about exact derivation of these expressions see[26]). In the rating task, the experimenter does not directly observe the decoded values $\hat{v}$ but the rating values $\check{v}$ on the physical bounded scale. The joint probability density $(\check{v}, v_0)$ on the rating scale is thus given by[26]

$$
\check{p}(\check{v}; v_0, \omega, \sigma_{\text{enc}}, \sigma_{\text{comp}}) = N(g^{-1}(\check{v}); E[\hat{v}|v_0], Var[\hat{v}|v_0] + \sigma_{\text{comp}}^2) \cdot p(v; \omega) \cdot (g^{-1}(\check{v}))',
\tag{25}
$$

where $g(\cdot)$ is the logistic function that provides a one-to-one mapping of the estimate $\hat{v}$ from the subjective to the physical scale on any given trial. In addition to internal noise $\sigma_{\text{enc}}$ in the coding of value, we also account for late noise in the decision stage (i.e., post-decoding noise), which may capture any unspecific forms of downstream noise occurring during the response process that are unrelated to the valuation process per se, which can include for instance late comparison noise. This external noise is represented as $\sigma_{\text{comp}}$ in the variance term of the normal distribution (Eq. (25)) which captures the random fluctuations. Thus a key feature of this inference model specification is that it allows separating noise in the encoding of values (which is by definition value specific) and value-unspecific noise that might be related to downstream processes.

In order to fit the efficient coding model to the rating data, we found the stimulus values $v_{1,...,M}$, parameters of the prior $\omega$, encoding noise $\sigma_{\text{enc}}$ and external noise $\sigma_{\text{comp}}$ that maximized the likelihood function $\check{p}(\check{v}; v_0, \omega, \sigma_{\text{enc}}, \sigma_{\text{comp}})$ (Eq. (25)) of the observed set of ratings for each participant under the constraint that $v_{1,...,M}$ is distributed following $p(v; \omega)$. Posterior inference of the parameters for this model can be conveniently performed via the Gibbs sampler.

Subsequently, we used the stimulus values $v_{1,...,M}$ and prior parameters $\omega$ fitted to the rating in order to predict choices in the two-alternative choice task. Please note that this strategy alleviates any concern for allowing an arbitrary choice of priors and likelihood functions in the choice models, given that the prior distribution is fully determined by out of sample data (i.e., the rating task) and the likelihood is fully constrained by the prior and our efficient coding specification. Based on this model, over many trials, the probability that an agent chooses an alternative with stimulus value $v_i$ over a second alternative with stimulus value $v_j$ is given by

$$
P(\hat{v}_i > \hat{v}_j | v_i, v_j) = \Phi\left(\frac{E[\hat{v}_i|v_i] - E[\hat{v}_j|v_j]}{\sqrt{Var[\hat{v}_i|v_i] + Var[\hat{v}_j|v_j] + \sigma_{\text{comp}}^2}}\right),
\tag{26}
$$

where $\Phi(\cdot)$ is the CDF of the standard normal distribution and the expressions for $E[]$ and $Var[]$ are given in Eqs. (23) and (24) (see above). Thus, the input values of the choice model are fully constrained by the efficient coding model based on the fits

to the rating data and therefore the choice model has only two free parameters, namely the resource noise of the encoder $\sigma_{enc}$ and the external noise $\sigma_{comp}$.

This choice model can be naturally extended to incorporate the effect of the attentional discounting parameter $\theta$

$$P(\hat{v}_i > \hat{v}_j | v_i, v_j) = \Phi\left(\frac{A_i E[\hat{v}_i | v_i] - A_j E[\hat{v}_j | v_j] + \theta(A_j E[\hat{v}_i | v_i] - A_i E[\hat{v}_j | v_j])}{\sqrt{Var[\hat{v}_i | v_i] + Var[\hat{v}_j | v_j] + \sigma_{comp}^2}}\right) \quad (27)$$

As before, to study the possible link between confidence reports and variable "$x$", we assume a linear relationship

$$x_n = x_{base} + \beta_s^x * c_n, \quad (28)$$

where $\beta_s^x$ is the effect of confidence $c_n$ on variable $x$ in a given trial $n$. The $\beta_s^x$ parameters are drawn from a population distribution $N(\beta^x, \sigma^x)$. Here, we list the uninformative priors of the population distributions

$$\begin{aligned}
\theta_{base} &\sim Uniform(-0.5, 1.5) \\
\sigma^{\theta_{base}} &\sim Uniform(10^{-10}, 2) \\
\beta^\theta &\sim Normal(0, 10^{-10}) \\
\sigma^\theta &\sim Uniform(10^{-10}, 5) \\
\sigma_{enc\_base} &\sim Uniform(10^{-10}, 5) \\
\sigma^{\sigma_{enc\_base}} &\sim Uniform(10^{-10}, 5) \\
\beta^{\sigma_{enc}} &\sim Uniform(-0.2, 0.5) \\
\sigma^{\sigma_{enc}} &\sim Uniform(10^{-10}, 1) \\
\sigma_{comp\_base} &\sim Uniform(10^{-10}, 5) \\
\sigma^{\sigma_{comp\_base}} &\sim Uniform(10^{-10}, 5) \\
\beta^{\sigma_{comp}} &\sim Normal(0, 10^{-10}) \\
\sigma^{\sigma_{comp}} &\sim Uniform(10^{-10}, 10)
\end{aligned} \quad (29)$$

**Behavioral analyses and statistics**. Rating variability in the data from our study was computed as the variance for each item across the rating phases 1 and 2. To investigate the influence of VD, rating variability, and reward value on the consistency of choices in each trial, we performed a hierarchical logistic mixed-effects regression of choices (defining consistent = 1, inconsistent = 0) on the above-mentioned regressors of interest, namely: VD, summed-variability (Var, defined as the sum of the two variances of the two food items presented in each trial), and the TV (defined as the sum of mean rating values of the two food items presented in each trial). All regressors of interest were included in the same model. Similarly, the influence of the same regressors on confidence was based on a hierarchical linear mixed-effects regression approach. All data analyses were performed in R 3.6.3, using RStudio and JAGS 4.3.0. BFs reported for these regressions were estimated based on the bridge sampling approach using the brms package in R[68].

All mixed-effects regressions in this study had varying subject-specific constants and slopes for the linear models, the joint modeling approach, and the RUMs parameter estimates, where the random effects parameter estimates are denoted in this work as $\beta$ unless otherwise specified. Posterior inference of the parameters in the hierarchical models was performed via the Gibbs sampler using the Markov Chain Monte Carlo (MCMC) technique implemented in JAGS, assuming flat priors for both the mean and the noise of the estimates. For each model, a total of 100,000 samples were drawn from an initial burn-in step and subsequently, a total of new 100,000 samples were drawn with three chains (each chain was derived based on a different random number generator engine, and each with a different seed). We applied a thinning of 100 to this final sample, thus resulting in a final set of 1000 samples for each parameter. We conducted Gelman-Rubin tests for each parameter to confirm the convergence of the chains. All latent variables in our Bayesian models had $\hat{R} < 1.05$, which suggests that all three chains converged to a target posterior distribution. We checked via visual inspection that the posterior population-level distributions of the final MCMC chains converged to our assumed parametrizations. For all random effects $\beta$ reported here, the reported value corresponds to the median of the posterior distribution, the ±values refer to 1 s.d. of the posterior distributions and the "$p$-values" reported for these regressions are not frequentist $p$-values but instead directly quantify the probability of the reported effect differing from zero. They were computed using the posterior population distributions estimated for each parameter and represent the portion of the density functions that lies above/below 0 (depending on the direction of the effect).

**Reporting summary**. Further information on research design is available in the Nature Research Reporting Summary linked to this article.

## Data availability
The data generated in this study have been deposited in the Open Science Framework database (https://doi.org/10.17605/OSF.IO/N7CUS)[69]. The data published by Folke et al.[4] is publicly available at (https://github.com/BDMLab/Folke_De_Martino_NHB_2016_Github) and (https://doi.org/10.6084/m9.figshare.3756144.v2). Source data are provided with this paper.

## Code availability
All code needed to replicate the results presented in this study has been made available at the Open Science Framework (https://doi.org/10.17605/OSF.IO/N7CUS)[69].

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

## Acknowledgements

We thank Hsin-Hung Li for their helpful comments and suggestions on the manuscript. This work was supported by a European Research Council (ERC) starting grant (ENTRAINER) to R.P. This project has received funding from the European Research Council (ERC) under the European Union's Horizon 2020 research and innovation program (grant agreement No. 758604).

## Author contributions

R.P. and M.G. conceived the original idea. H.A. performed the experimental studies and analyzed data. M.G. pre-processed eye-tracking data and contributed to manuscript writing. J.B. carried out data analyses, implemented computational models, and wrote the manuscript. M.G., J.B., and R.P. discussed and interpreted the results and developed the concepts. R.P. contributed to data analyses, manuscript writing, and acquired funding.

## Competing interests

The authors declare no competing interests.
