## [Peer Review File · Nature Communications]

Sources of confidence in value-based choiceREVIEWER COMMENTS

Reviewer #1 (Remarks to the Author):

In this paper Brus and colleagues introduce a series of new computational ideas to pin down the relation between decision confidence in value-based choice, noise and attention. They test the predictions of a series of models on a new dataset and validate them on an independent (published) dataset. I must say that I really liked this paper! The topic is extremely timely and the approach is novel and refreshing. The authors have also been very scrupulous with their analyses, model implementations and out of sample validation; therefore all the main conclusions are robustly supported. The paper is overall well written however sometime I feel that the narrative thread is a bit lost. Also is not super clear from the abstract/introduction which is the central idea of the manuscript. I am giving below some more specific suggestions that I hope the authors will find useful in their revision:

Main point

There are many different modelling approaches presented in the result section but is not always clear the link between them. Also is sometime unclear how a new modelling approach is a follow-up/improvement of the previous approach or just a separate/independent approach.

Let me summarise what I understood: initially the authors present a classic balance of evidence model (the one they call heuristic confidence) implemented through a GLAM (therefore taking into account the effect of attention) and compare it against a 'normative confidence' model, in which they estimate (optimally) the position of the losing accumulator (Fig 1). Then in the Fig 2 they show the results of a joint covariance approach that allow the comparison of the 2 models (presented in fig 1) also testing the hypothesis that trials by trial fluctuations in attention affect the estimation of confidence locally. Here is when the story get a bit confusing for me. The authors in this new section shift gears and present the RUM model. However, here we are not talking anymore about choice data but rating data (note this should be made much clearer). Also is this new model compatible with the ones used for choice data? It seems so since this is on a completely different type of data. However, the authors introduce this model saying that improve on the previous approach (Fig 2) that is computationally demanding (BTW in real term is not so demanding — I run routinely these hierarchical Bayesian models on my laptop). But here I am confused, the authors never implemented a similar graphical model for rating data, am I correct? Also what this modelling of the rating data add to their argument?(beside it is intrinsic value in describing the data well). Maybe if the behavioural results in figure one are split in choice and rating each one followed by the relevant models this could become clear? This is just a thought but I am sure the authors can find also other way to make this separation more clear

Finally (and this is the most confusing thing for me) the authors in Fig 5 introduce yet *another* completely different modelling approach based on their recent work (the efficient coding model). However now they can test this model only one 1 of the 2 dataset they have used so far (since Folke et al does not measure rating multiple times). At this point becomes a bit confusing to see how all these very different approaches come all together to make one coherent narrative. I feel that this last model, add more confusion/ cognitive demand for the reader, while does not add much to the manuscript. If the authors agree, I see this fig 5 more fit for the supplemental material. This should help the paper to become more streamlined.

Minor points

— If I understand correctly, the idea that the authors call ‘heuristic confidence’ is nothing else than what in the field of perceptual decision making is known as ‘balance of evidence’ an idea proposed by Douglas Vickers in 1979. This is in itself a re-formulation of a simple detection-theory idea in the context of dynamic model such as race models (a big class of model that include the GLAM model). There is no reference to this in the manuscript. I think it would be good to highlight this since it will make it easier to put these results into context. In particular for an audience more attuned to perceptual DM.

— Could the author elaborate on the fact (page 4, 2nd column, 1st paragraph) that only the normative model predicts that higher attentional effort (theta parameter) leads to higher confidence reports (I assume we are talking here of the TV effect). I am a bit puzzled by this since, in my understanding, the heuristic model (that I believe is the classic balance of evidence) without attentional modulation does not predict the total value effect. This instead appears only when the GLAM allows for attention allocation to break the symmetry in the accumulation process for the 2 options (see also Sepulveda et al 2020).

— Figure 4 is a bit confusing for me since everything is mirrored between the 2 studies (that is a good thing) but then panels A and E are not. In one case it is a schematic of the RUM model and in another case it is the task. This also relates to the other point of making it clearer between modelling of choice vs rating data. Please make it consistent across figures since it would be helpful for many readers.

— Code/data availability: I could not find any repositories/GitHub links where data and codes are made freely available. I would strongly encourage the authors to do so (without the need to contact the authors). As they have themselves demonstrated in this manuscript the fact that all the data/codes for the study of Folke et al were publicly available has made it easy and possible for them to use such datasets to test their hypotheses more robustly. It would be great if future studies can also capitalise on these new data and build up on these models. It will make the paper more relevant/cited in the field and (most importantly) will allow for swift scientific progress.

I hope that the authors find my comments useful
best
Benedetto De Martino

Reviewer #2 (Remarks to the Author):

Brus et al. make an important contribution to uncovering the mechanisms behind confidence judgements in value-based decisions, which are intrinsically different from the perceptual case where the stimuli and the evidence can be objectively controlled. The authors use computational modeling and a task they recently developed (that allows to explicitly compute an index of value uncertainty for each option presented), to disentangle the contribution of attentional effort and noise within the decision process on choice and confidence. The main findings are that economic choice and confidence are affected by different processes/variables. Choice is affected by fluctuations in value precision, while confidence reflects endogenous attentional effort toward the option compared and noise in the comparison process. I feel this paper opens new perspectives to the field and is likely to have a lasting impact. It also made me think more about the multiplexed nature of

confidence. I have some comments and concerns, as outlined below.

While the difference in subjective value between options can be taken as the strength of evidence, the total value is closer to the amount of positive evidence often discussed in the perceptual literature (e.g., Zylberberg et al 2012, Koizumi et al 2015).

From these studies (but other too), it appears that confidence is not as much affected by the variability of evidence, or the difference in evidence between the two options, but rather is strongly affected by the total amount of evidence in favor of the chosen option. In this case, by the total amount of value (since the total value here is taken as the sum across options, it correlates highly with the total value of the chosen option).

An analysis separating the contribution of the value of the chosen and unchosen options would probably result in the same outcome.

Thus, it is not entirely surprising that confidence strongly correlates with the total value and not with the variability in value. At least, this results seems expected given the findings in perceptual decision making.

Putting these results in context would strengthen the manuscript - particularly for the authors' claim that in value-based decisions confidence is computed on different terms.

Relatedly, the fact that uncertainty about value (signal) has no/little influence on confidence is a great finding. But it is true that it has also been similarly shown in perceptual decisions where - as mentioned above, confidence appears to be mainly driven by the amount of positive evidence. Though, in those cases, this was not tested explicitly as done in this study.

In the manuscript the authors state (e.g., intro, but in the discussion too) that one key difference between value-based decision and classical perceptual decision paradigms is that value-based choice tasks usually entail two or more alternatives presented simultaneously, such that participants can shift their gaze between options. There are several studies in perceptual decision-making that used this kind of paradigm, such as Zylberberg et al 2012 (already cited), Rahnev et al. 2011, Roualt et al. 2018 and others. Furthermore, in many perceptual tasks with one stimulus presented at a time, the stimulus contains information relevant to both choice option (e.g., in random dot motion, left and right) (Zylberberg et al. 2012, Koizumi et al. 2015). Thus, these paradigms in perception seem relatively comparable in nature, at a conceptual level, with value-based decision making tasks.

It would be useful if in the methods the hyper-parameters of each model were listed.

In the authors words, the fact that confidence reflects the degree of balance and cognitive effort with which the choice alternatives have been compared:

To what extent does this depend on the way the question is formulated? That is, on how participants are asked to rate their confidence?

Related to confidence ratings - it seems that in the main experiment (as opposed to Folke et al.) participants used confidence in a rather restricted manner across value-difference levels (with little differences). Although the effects and particularly the interaction correct/error trials are significant, I wonder what is the reason for this difference. Did participants use the full confidence scale (it would appear so based on the RT and TV plots)? Did participants use the full like rating scale for the value measure? If not, these might be problematic for the interpretation of some of the results. It would be good to see this point clarified, with additional visualization of the raw data.

The finding that attentional effort is positively associated with confidence is exciting and novel (to me), and as the authors state - indicates that agents have some form of insight into the evidence accumulation process (since attention effort will reflect how balanced is their information seeking across options). It is particularly interesting because in the perceptual decision making literature attention towards a stimulus appears to lead to lower confidence judgements (e.g., Rahnev et al. 2011, Morales et al. 2015). Now, these two findings are not opposed, but it would be interesting to see the authors discussing their findings in this context too.

In the methods the authors write that trials where participants had not fixated on every option available at least once were excluded to ensure participants deliberated between options. Not fixating on one option might not necessarily mean that participants did not deliberate, because one can get the gist of a scene without focusing on each part.

Please indicate how many trials were thus excluded for each participants. If the number is high, please replicate the main results using all trials (I think a recent study by Sepulveda et al. eLife 2020 where GLAM was used, all trials were included).

There is a rich history in perceptual decision making that confidence reports do not follow a Bayesian optimal process - i.e., that confidence can be dissociated from accuracy or the strength of evidence (e.g., DePaulo et al 1997, Komura et al 2012, Koizumi et al 2015, Samaha et al. 2017, Maniscalco et al. 2021). It is true that in the animal literature studies have generally taken a Bayesian/statistical approach, but when reading the manuscript the impression was that in previous studies of confidence (esp. in perceptual decisions), confidence has been so far assumed Bayesian optimal and the authors' study was the first to show evidence against it.

It was not immediately clear that the results in Fig3 were from the normative model.

It seems Fig4a-b, 4e-f were not introduced in the text. Introducing the type of betting in the Folks task in Fig4e is kind of strange, since the data was already used in Fig3.

There are many references to methods in the text that make understanding the paper/approach difficult at times (going back and forth).

Since eye-tracking data play a major role in the modeling, some basic / summary analysis / visualizations could be reported in supplementary.

Section "Confidence reports are not related to trial-to-trial fluctuations in reward encoding" While earlier on the term value was used, here the term reward is used. It might be obvious for some readers, but I feel it will be confusing to others (it was to me). I suggest using 'value' throughout since in this context reward feels intangible.

When comparing two correlations that imply a difference in effect this should be tested explicitly (fig 1d, though this is done elsewhere in the manuscript).

In figure 2e it would be useful if more information were given (e.g. in the legend): how are these regressions results obtained? What do bars and error bars represent?

The way figures are referenced in the text is not always clear. Some panels are not mentioned/introduced at all in the text.

References:

- Zylberberg, Ariel, Pablo Barttfeld, and Mariano Sigman. 2012. "The Construction of Confidence in a Perceptual Decision." *Frontiers in Integrative Neuroscience* 6. <https://doi.org/10.3389/fnint.2012.00079>.
- Rahnev, Dobromir, Brian Maniscalco, Tashina Graves, Elliott Huang, Floris P. de Lange, and Hakwan Lau. 2011. "Attention Induces Conservative Subjective Biases in Visual Perception." *Nature Neuroscience* 14 (12): 1513–15.
- Koizumi, Ai, Brian Maniscalco, and Hakwan Lau. 2015. "Does Perceptual Confidence Facilitate Cognitive Control?" *Attention, Perception & Psychophysics* 77 (4): 1295–1306.
- Samaha, Jason, Luca Iemi, and Bradley R. Postle. 2017. "Prestimulus Alpha-Band Power Biases Visual Discrimination Confidence, but Not Accuracy." *Consciousness and Cognition* 54 (September): 47–55.
- Rouault, Marion, Tricia Seow, Claire M. Gillan, and Stephen M. Fleming. 2018. "Psychiatric Symptom Dimensions Are Associated With Dissociable Shifts in Metacognition but Not Task Performance." *Biological Psychiatry* 84 (6): 443–51.
- Morales, Jorge, Guillermo Solovey, Brian Maniscalco, Dobromir Rahnev, Floris Lange, and Hakwan Lau. 2015. "Low Attention Impairs Optimal Incorporation of Prior Knowledge in Perceptual Decisions." *Attention, Perception & Psychophysics*. <https://doi.org/10.3758/s13414-015-0897-2>.
- Komura, Yutaka, Akihiko Nikkuni, Noriko Hirashima, Teppei Uetake, and Aki Miyamoto. 2013. "Responses of Pulvinar Neurons Reflect a Subject's Confidence in Visual Categorization." *Nature Neuroscience* 16 (6): 749–55.
- DePaulo, B. M., Kelly, C., Harris, C., Lindsay, J. J. & Laura, M. The accuracy-confidence correlation in the detection of deception. *Pers. Soc. Psychol. Rev.* 1, 346–357 (1997).
- Maniscalco, Brian, Brian Odegaard, Piercesare Grimaldi, Seong Hah Cho, Michele A. Basso, Hakwan Lau, and Megan A. K. Peters. 2021. "Tuned Inhibition in Perceptual Decision-Making Circuits Can Explain Seemingly Suboptimal Confidence Behavior." *PLoS Computational Biology* 17 (3): e1008779.
- Sepulveda, Pradyumna, Marius Usher, Ned Davies, Amy A. Benson, Pietro Ortoleva, and Benedetto De Martino. 2020. "Visual Attention Modulates the Integration of Goal-Relevant Evidence and Not Value." *eLife* 9 (November).

Thank you very much for reviewing our manuscript entitled “*Sources of confidence in value-based choice*”. We thank you for your positive and constructive feedback and your well-structured comments, which were very useful in guiding our revision and substantially improved the paper. We hope you agree that our revised manuscript will have a considerable impact on the field and is suitable for publication in Nature Communications.

In the following, we provide a point-by-point revision list addressing all the reviewers’ comments. Reviewers’ comments are in italics and (our point-by-point clarifications are in normal font). Changes in the manuscript text are highlighted in blue font.

Reviewers' comments

Reviewer #1

In this paper Brus and colleagues introduce a series of new computational ideas to pin down the relation between decision confidence in value-based choice, noise and attention. They test the predictions of a series of models on a new dataset and validate them on an independent (published) dataset. I must say that I really liked this paper! The topic is extremely timely and the approach is novel and refreshing. The authors have also been very scrupulous with their analyses, model implementations and out of sample validation; therefore all the main conclusions are robustly supported. The paper is overall well written however sometime I feel that the narrative thread is a bit lost. Also is not super clear from the abstract/introduction which is the central idea of the manuscript. I am giving below some more specific suggestions that I hope the authors will find useful in their revision:

Main Point

There are many different modelling approaches presented in the result section but is not always clear the link between them. Also is sometime unclear how a new modelling approach is a follow-up/improvement of the previous approach or just a separate/independent approach.

Let me summarise what I understood: initially the authors present a classic balance of evidence model (the one they call heuristic confidence) implemented through a GLAM (therefore taking into account the effect of attention) and compare it against a 'normative confidence' model, in which they estimate (optimally) the position of the losing accumulator (Fig 1). Then in the Fig 2 they show the results of a joint covariance approach that allow the comparison of the 2 models (presented in fig 1) also testing the hypothesis that trials by trial fluctuations in attention affect the estimation of confidence locally. Here is when the story get a bit confusing for me. The authors in this new section shift gears and present the RUM model. However, here we are not talking anymore about choice data but rating data (note this should be made much clearer). Also is this new model compatible with the ones used for choice data? It seems so since this is on a completely different type of data. However, the authors introduce this model saying that improve on the previous approach (Fig 2) that is computationally demanding (BTW in real term is not so demanding — I run routinely these hierarchical Bayesian models on my laptop). But here I am confused, the authors never implemented a similar graphical model for rating data, am I correct? Also what this modelling of the rating data add to their argument?(beside it is intrinsic value in describing the data well). Maybe if the behavioural results in figure one are split in choice and rating each one followed by the relevant models this could become clear? This is just a thought but I am sure the authors can find also other way to make this separation more clear

*Finally (and this is the most confusing thing for me) the authors in Fig 5 introduce yet *another* completely different modelling approach based on their recent work (the efficient coding model). However now they can test this model only one 1 of the 2 dataset they have used so far (since Folke et al does not measure rating multiple times). At this point becomes a bit confusing to see how all these very different approaches come all together to make one coherent narrative. I feel*

that this last model, add more confusion/ cognitive demand for the reader, while does not add much to the manuscript. If the authors agree, I see this fig 5 more fit for the supplemental material. This should help the paper to become more streamlined.

Resp: Thank you for this thoughtful comment which motivated us to improve the outline and description of the different approaches throughout the manuscript. We agree with the reviewer that the paper would benefit from an early introduction of the goals of the computational models and how and why we transition from one approach to the other. As we describe in more detail below, we have added a short paragraph outlining and motivating the different models and clarified how the models are related to each other (page 2, right column in revised manuscript).

We now turn to the specific questions of the reviewer in this comment.

RUM model. We apologize for the misunderstanding generated by our description regarding the RUM model. The RUM is also concerned with choice data. The advantage of the RUM is (1) that it requires shorter computational time, (2) allows us to validate the conclusions of the initial results in the absence of reaction time data, and (3) this approach can be parsimoniously extended to formally incorporate the statistics of the environment, which in turn allows to disentangle the influence of noise on confidence at different stages of reward and decision processing such as value encoding noise and downstream noise in the comparison process.

We agree with the reviewer that nowadays computational times are not an issue for this class of models when implemented based on standard sequential sampling model specifications. However, a key difference between the approach adopted in our work and the standard specifications is that the “joint-modelling approach” relies on the estimation of covariance matrices that are computationally demanding (in particular for models with hierarchical Bayesian structures in which each element of the matrix is modelled as a random variable). For the case of this model applied to our data it took 2.4 days for all chains to finish with 1,000 samples per chain (thinning 100, adapt plus burnin 100,000, with parallel chains, i.e., a core per chain). Moreover, access to reaction time data and specification of dynamical models is not always possible in some datasets and studies of psychology and economics (nevertheless, we emphasize that whenever possible, RT and decision data should jointly be used as they provide complementary information underlying decision processes, and we highlight this fact in our work). Therefore, we believe that it was important to demonstrate that access to co-fluctuations between confidence reports and latent variables of decision models can also be accessed without the need of having RT information, a result that might be appealing for some of our Psychology and Behavioral Economic peers. Additionally, we show that this family of “RT-free models” can be parsimoniously extended to include the statistics of the environment to disentangle different sources of noise (e.g., encoding noise from comparison noise as is done in the efficient coding model, which in our specification in the original article does not formally incorporate RTs in the normative solution). We now make the motivation and rationale for the implementation of the RUM model clearer in the revised manuscript (page 6, right column). Additionally, we clarify how the RUM can be naturally extended to the efficient coding formulation (page 9, right column).

Efficient coding model. We have considered the reviewer’s advice to move fig 5 and the corresponding results on the efficient coding model to the supplemental material, however in our

eyes (and reviewer 2 seems to agree), the result that encoding noise has no effect on confidence is a key finding of the current work. As we mentioned above, in the revised version of the manuscript we now try to motivate the utility of the efficient coding model and the importance of the findings resulting from the application of this model.

Abstract / Introduction. We agree with the reviewer that the abstract and parts of the introduction in the original submission could contain more specific information about the central idea of the article. We have now modified the abstract such that it reflects the central points of our work (note that we had to substantially reduce the length of the abstract due to the word-count limit of 150 words, but we tried our best to incorporate all key elements). Also, we have modified parts of the introduction to better reflect key aspects that are studied in our work. Moreover, we added an outline of the modelling approach in the first part of the results section where we provide a brief roadmap of the analyses and modelling approach implemented in our work such that the aims and key aspects that are studied in our work are better emphasized.

Minor Points

— *If I understand correctly, the idea that the authors call 'heuristic confidence' is nothing else that what in the field of perceptual decision making is known as 'balance of evidence' an idea proposed by Douglas Vickers in 1979. This is in itself a re-formulation of a simple detection-theory idea in the context of dynamic model such as race models (a big class of model that include the GLAM model). There is not reference to this in the manuscript. I think would be good to highlight this since it will make easier to put these result into context. In particular for an audience more attuned to perceptual DM.*

Resp: The reviewer's observation is correct and we agree that this information should be made more explicit. We now clarify that the heuristic model of confidence is based on the classic 'balance of evidence' approach and we now include the reference to Douglas Vickers book from 1979 (page 4, left column, 4th paragraph).

“... The heuristic confidence model is based on the classic "balance of evidence" approach (D.Vickers 1979). In brief, this approach proposes that the observer is conceived as accumulating successive differences between momentarily registered noisy values of two alternatives i and j . As soon as one of the total accumulated differences reaches a decision bound B , the observer makes a decision in favor of the winner alternative. A workable definition of confidence is based on the "balance of evidence", that is, on the difference between the total evidence of the two accumulators $e_i(t)$ and $e_j(t)$ at decision time t . The closer the particle of the loser alternative is to the decision bound, the lower the confidence of the observer (DeMartino 2013, Sepulveda 2020). ... ”

— *Could the author elaborate on the fact (page 4, 2nd column, 1 paragraph) that only the normative model predicts that higher attentional effort (theta parameter) leads to higher confidence report (I assume we are talking here of the TV effect). I am a bit puzzled by this since, in my understanding, the heuristic model (that I believe is the classic balance of evidence) without attentional modulation does not predict the total value effect. This instead appears only*

when the GLAM allows for attention allocation to break the symmetry in the accumulation process for the 2 options (see also Sepulveda et al 2020).

Resp: We agree that this point is important to clarify. The answer to this question comes in two parts:

On the one hand, the fact that higher attentional effort does not lead to higher confidence in the heuristic model can be intuitively understood when we consider that an item that is looked at longer will be more likely to be chosen. Higher attentional effort translates to better incorporation of the item that is not looked at into the decision process. Therefore, higher attentional effort will result in more accumulated evidence for the unchosen item, and ultimately in less confidence. This can be made explicit with a parsimonious calculation like in Sepulveda et al 2020. We investigate two scenarios that explain how higher theta leads to lower confidence. Scenario 1 (low theta): We choose input values $in_1 = 1$, $in_2 = 2$, and a theta of 0.3 and we assume that the item with higher value is looked at more. Following the example of Sepulveda, we can simplify the decision process and state the momentary evidence for option 1 = $in_1 * \theta = 0.3$ and for option 2 = $in_2 = 2$. Which leads to a difference in evidence of 1.7. In scenario 2 (high theta): the input values in_1 and in_2 are the same as in scenario 1, but $\theta = 0.8$, which leads to option 1 = $in_1 * \theta = 0.8$ and option 2 = $in_2 = 2$, which comes down to a difference in evidence of 1.2. Thus, in the higher theta scenario there is a smaller difference in evidence which leads to lower confidence. On the other hand, the normative model rationalizes the influence of attentional effort via the average log-odds estimation. In general, with higher attentional effort both options are compared more fairly which leads to a higher chance of choosing the correct item, and this is further boosted under the assumption of variability in both the latent attentional effort parameter (with $\theta < 1$) and variability in gaze dynamics).

Second, we investigate two scenarios that explain how higher total value leads to higher confidence, while keeping their absolute value differences identical. Scenario 1 (low TV): We choose input values $in_1 = 1$, $in_2 = 2$, and a theta of 0.3 and we assume that the item with higher value is looked at more. Again, we simplify the decision process and state the momentary evidence for option 1 = $in_1 * \theta = 0.3$ and for option 2 = $in_2 = 2$. Which leads to a difference in evidence of 1.7. In scenario 2 (high TV): $in_1 = 4$, $in_2 = 5$, which leads to option 1 = $in_1 * \theta = 1.2$ and option 2 = $in_2 = 5$, which comes down to a difference in evidence of 3.8. Thus, in the higher total value scenario there is a greater difference in evidence which leads to higher confidence. Interestingly, total value affects confidence reports in both models in similar ways (compare the standardized effects of total value in Figure 2e).

Taken together, these examples and observations give insight to how attentional effort induces distinct influences in the heuristic model and normative model, and how total value induces similar influences in both models. We are happy to provide such example in a Supplementary Note if the reviewer and the editor consider that it may help to clarify how these variables influence confidence in each model.

— *Figure 4 is a bit confusing for me since everything is mirrored between the 2 studies (that is a good thing) but then panels A and E are not. In one case is a schematic of the RUM model and in another case is the task. This relates also to the other point of making more clear between modelling of choice vs rating data. Please make it consistent across figures since it would be helpful for many readers.*

Resp: Thank you for this suggestion. Reviewer 2 also commented on figure 4, correctly noting that we use data of the Folke et al. paper already in figure 3. Therefore, addressing the observation of both reviewers we decided to move panel 4e to figure 3 in the revised manuscript. In this way comparison of the information across both datasets is clearer in both figures (see new figures 3 and 4 in the revised manuscript).

— *Code/data availability: I could not find any repositories/GitHub links where data and codes are made freely available. I would strongly encourage the authors to do so (without the need to contact the authors). As they have themselves demonstrated in this manuscript the fact that all the data/codes for the study of Folke et al were publicly available has made easy and possible for them to use such dataset testing their hypotheses more robustly. It would be great if future studies can also capitalise on these new data and build up on these models. It will make the paper more relevant/cited in the field and (most importantly) will be allow swift scientific progress.*

Resp: We completely agree with the reviewer on this point. We will make the data as well as the essential code publicly available upon publication of the article. Code and data will be released in the Open Science Framework platform, here is the link:

https://osf.io/n7cus/?view_only=da41dfe1bf7149fab0d4c1f4690644cd

Reviewer #2

Ile Brus et al. make an important contribution to uncovering the mechanisms behind confidence judgements in value-based decisions, which are intrinsically different from the perceptual case where the stimuli and the evidence can be objectively controlled. The authors use computational modeling and a task they recently developed (that allows to explicitly compute an index of value uncertainty for each option presented), to disentangle the contribution of attentional effort and noise within the decision process on choice and confidence. The main findings are that economic choice and confidence are affected by different processes/variables. Choice is affected by fluctuations in value precision, while confidence reflects endogenous attentional effort toward the option compared and noise in the comparison process. I feel this paper opens new perspectives to the field and is likely to have a lasting impact. It also made me think more about the multiplexed nature of confidence. I have some comments and concerns, as outlined below.

Reviewer comments

1. While the difference in subjective value between options can be taken as the strength of evidence, the total value is closer to the amount of positive evidence often discussed in the perceptual literature (e.g., Zylberberg et al 2012, Koizumi et al 2015).

From these studies (but other too), it appears that confidence is not as much affected by the variability of evidence, or the difference in evidence between the two options, but rather is strongly affected by the total amount of evidence in favor of the chosen option. In this case, by the total amount of value (since the total value here is taken as the sum across options, it correlates highly with the total value of the chosen option).

An analysis separating the contribution of the value of the chosen and unchosen options would probably result in the same outcome. Thus, it is not entirely surprising that confidence strongly correlates with the total value and not with the variability in value. At least, this results seems expected given the findings in perceptual decision making. Putting these results in context would strengthen the manuscript - particularly for the authors' claim that in value-based decisions confidence is computed on different terms.

Relatedly, the fact that uncertainty about value (signal) has no/little influence on confidence is a great finding. But it is true that it has also been similarly shown in perceptual decisions where - as mentioned above, confidence appears to be mainly driven by the amount of positive evidence. Though, in those cases, this was not tested explicitly as done in this study.

Resp: We thank the reviewer for this thoughtful comment and for pointing us to these findings in the perceptual decision-making literature. We agree that our results on total value might be in line with earlier results in the perceptual decision domain, which show that positive evidence influences confidence more strongly.

We investigated this aspect more closely in our data by repeating the model-free analyses trying to separate the contribution of the value of the chosen and unchosen options, see figure 1 below. Note that in a single regression model, value difference (VD), total value (TV), positive evidence (PE) and negative evidence (NE) cannot be used as factors as there will be collinearities. Therefore, we have compared three models using the following factors to explain choice and confidence as dependent variables

Model 1: VD + TV + Var (as in the original submission)

Model 2: VD + PE + Var

Model 3: VD + NE + Var

Specifying models in this way allows us to study whether the effects of Variability are present in all choice models and absent in all confidence models. Additionally, it allows us to compare the influence of PE and NE based on their contributions in the respective models while always controlling for both VD and Var. Supporting our main argument, we found that the effects of Variability are indeed present in all choice models and absent in all confidence models.

Additionally, and in line with the reviewer's observation, we found that the effects of PE are slightly higher than the effects of NE ($\Delta\beta = 0.10 \pm 0.08$, $P_{\text{MCMC}} = 0.11$; note that in all regressions the factors are standardized relative to the data of the whole population thus allowing to quantify the relative contributions of these two factors). However, we caution that due to relatively high correlation between total value, positive evidence and negative evidence in our study, we cannot conclude whether the true underlying factor here is total value or positive evidence. To answer this question a future study should be designed that carefully orthogonalizes these factors. Nonetheless we have added the results presented in Figure 1 (below) as a supplementary figure (new Supplementary Figure 11 in the revised manuscript). Additionally, as suggested by the reviewer, we have now added a paragraph in the discussion where we put our results in context with the perceptual decision-making literature (p. 10, right column)

“... For instance, we find that total value has no effect on choice consistency but does affect confidence ratings. While initially surprising, this result appears in line with observations from perceptual decision-making studies showing that sources of evidence supporting the correct stimulus identification response (i.e., positive evidence) and sources of evidence that support the alternative response (negative evidence) both influence choice accuracy, but positive evidence affects confidence more strongly (Zylberberg2012, Koizumi2015). However, based on the standard specifications of value-based decisions adopted in our study, we cannot ascertain whether the true factor underlying this phenomenon is total value or positive evidence as by design these two factors are highly correlated (see Supplementary Fig. 11). ...”.

Fig. 1: a,b) Standardized estimates of three multiple regression analyses of choice consistency and confidence. Total value, positive evidence and negative evidence are used as explanatory variables in separate regressions, while controlling for value difference and variability. **c)** Investigating the difference of the beta values of PE and NE on confidence we find that there is a trend for the effect of PE to be larger, however this effect is not significant (11% of the density is below 0).

2. In the manuscript the authors state (e.g., intro, but in the discussion too) that one key difference between value-based decision and classical perceptual decision paradigms is that value-based choice tasks usually entail two or more alternatives presented simultaneously, such that participants can shift their gaze between options. There are several studies in perceptual decision-making that used this kind of paradigm, such as Zylberberg et al 2012 (already cited), Rahnev et al. 2011, Roualt et al. 2018 and others. Furthermore, in many perceptual tasks with one stimulus presented at a time, the stimulus contains information relevant to both choice option (e.g., in random dot motion, left and right) (Zylberberg et al. 2012, Koizumi et al. 2015). Thus, these paradigms in perception seem relatively comparable in nature, at a conceptual level, with value-based decision making tasks.

Resp: We thank with the reviewer for this comment and agree that the way in which we phrased this aspect of value-based decisions in the introduction may have led to confusion and misinterpretation. We have changed the relevant part of the text (page 1, right column, 3rd paragraph), to clarify that information obtained from gaze shifting behaviour is key to the analyses of value-based choice paradigms typically implemented in economic decision-making

studies – which is not structurally different from behavioural paradigms of perceptual decision-making.

Another key aspect to consider is that value-based choice tasks usually entail two or more alternatives for choice situated at different spatial locations of the visual field. That is, decision makers foveate - often repeatedly via changes in eye-fixation - to one of the choice options at a time, thereby gathering evidence from each alternative (Krajbich 2010). Seminal studies in the attention literature clearly indicate substantial influences of reward value in biasing attention with direct consequences on choice processes (Peck 2009, Anderson 2011, Grueschow 2015). Thus, an unresolved question is whether observers have the capability to introspect about their endogenous attentional states of the decision processes, and whether these internal signals are used to inform post-decision confidence reports.

As for the discussion, here we are not trying to argue that the seemingly opposing results in our paper and the paper of Castanon et al. (Castanon 2019) originate from gaze shifting behaviour. Instead, we argue that these seemingly conflicting results are actually rooted in different definitions of encoding (early) and comparison (late) noise. In the cited paper late noise is influencing the estimation of the mean angle of multiple gratings and increases with increasing spread of orientation of the individual angles. Encoding noise is caused by the contrast with which these individual gratings are presented. We argue that in our study encoding noise is very similar to the late noise mentioned above, since value estimation is inherently an integrative process taking into account multiple attributes of the presented item and different discrete components such as memories and emotions. What we call comparison noise is the noise introduced by comparing the values of the two items and is something not studied by Castanon et al. We have updated the relevant paragraph discussing this topic (page 11, left column). We now prevent any misunderstanding that we may claim this is a fundamental difference between the value-based and perceptual decision making domain, but rather a difference in the definition of encoding noise.

"Regarding our established dissociation between encoding and comparison noise, at first sight this result seems at odds with recently reported results by Castanon et al. (Castanon 2019). It was found that humans disregard downstream integration noise, while using encoding noise to form beliefs about confidence. Here, we argue that the apparent difference between the influence of encoding noise on confidence in our study and the study of Castanon et al. has its basis in the definition of encoding noise and the characteristics of the task. Castanon et al. use a categorization task where they present multiple tilted gratings in a circular array and vary either the contrast level of the gratings (resulting in different levels of encoding noise) or the variability of the gratings' orientations (resulting in different levels of integration noise, which they define as late noise). In their task subjects are asked to categorize the average orientation as tilted clockwise or counter-clockwise. First, we argue that what the authors of this work define as integration noise, could be very much related to what we define as encoding noise in our reward task. It has been suggested and is commonly accepted that reward value is formed via the (integration) of different discrete components (e.g., memories and emotions) associated to the physical features of the choice alternatives (Shadlen 2016). Therefore, it is well possible that trial-to-trial fluctuations in reward value estimations and its degree of variation is related to the definitions of integration noise in the above-mentioned categorization tasks. This would be in principle congruent with a previous study in the perceptual decision-making domain, demonstrating that observers underestimate the variance of orientation noise, which leads to distorted confidence reports (Zylberberg 2014). Second, in our task, observers (integrate) information of two distinct choice alternatives thus

resulting in two distinct value estimates that need to be compared via down-stream circuits. However, typically in categorization tasks observers integrate multiple cues and generate a single value estimate to be categorized. Therefore, we argue that perceptual decision tasks of the kind discussed above would resemble the characteristics of our task only in the event where feature integration of two distinct feature estimates takes place simultaneously (e.g., the comparison of two distinct arrays of tilted gratings) and then compared according to an abstract decision rule (e.g., which of the mean estimates is closer to cardinal). It will be interesting to investigate in future work whether a perceptual task designed in this way would also reveal that down-stream comparison noise heavily influences confidence reports. ...”.

3. It would be useful if in the methods the hyper-parameters of each model were listed.

Resp: On page 14, right column, page 15, right column and page 16, right column, we have now listed the hyper-parameters alongside their parameter spaces.

4. In the authors words, the fact that confidence reflects the degree of balance and cognitive effort with which the choice alternatives have been compared:

To what extent does this depend on the way the question is formulated? That is, on how participants are asked to rate their confidence?

Resp: This is an interesting question that is difficult to answer in an objective manner. If our study and the study of Folke et al. had used different formulations of the confidence question, we could have directly answered this question. However, our study and the study from Folke et al. use a very similar formulation and confidence rating scale (which does not greatly differ from formulations in previous studies (e.g., DeMartino et al., 2013)). However, other studies sometimes ask participants to indicate their confidence on a discrete scale by pressing one out of four or five buttons (e.g. Sanders et al., 2016), instead of a continuous scale. In these discrete rating studies, a score of 1 to indicates a random guess and button 5 to indicate high confidence. We feel that such an approach would not necessarily have been beneficial for our study, if anything we feel a continuous scale gives the participant more freedom to indicate their confidence. Irrespective of these considerations, we took a closer look at the interindividual variability in confidence reports and it appears that the rating scale was used in a nearly consistent manner across participants (see the response to comment #5, below).

5. Related to confidence ratings - it seems that in the main experiment (as opposed to Folke et al.) participants used confidence in a rather restricted manner across value-difference levels (with little differences). Although the effects and particularly the interaction correct/error trials are significant, I wonder what is the reason for this difference. Did participants use the full confidence scale (it would appear so based on the RT and TV plots)? Did participants use the

full like rating scale for the value measure? If not, these might be problematic for the interpretation of some of the results. It would be good to see this point clarified, with additional visualization of the raw data.

Resp: To address this comment we followed the reviewer's advice and investigated the distribution in the usage of the liking and confidence rating scales (see figure 2, below).

On the one hand, the use of the liking rating distributions shows a pattern similar to the one observed in previous studies (e.g., Polania et al., 2019) where the high degree of variability across participants in the use of the rating scale may be related to the "liking priors" that each participant has on the food items used in our experiment.

On the other hand, the use of the confidence ratings reveals less variability across participants with relative increase of confidence ratings towards higher values. This is expected given that decision accuracy is about 75% in our task and the proportion of correct responses is much higher than the incorrect responses. Therefore, the use of the confidence rating scale (according to the statistical definition of confidence) should show that the higher end of the scale is more often used.

The analyses revealed that two participants indicated extreme high degrees of confidence most of the time, see figure 2. We investigated whether removing these two participants has a major impact on the main conclusions of our work. We found no major qualitative and quantitative differences when we excluded these two participants. Taken together, we believe that it is safe to conclude that generally the participants made use of the rating scale similarly and as intended (except for the two outliers indicated in the figure).

Fig. 2: a) Usage of the preference and confidence rating scale. Colored lines correspond to individual participants, black lines and gray shadings are aggregated data over all subjects. Participants use the full preference rating scale and mostly the positive part of the confidence rating scale. There are two outlier participants consistently reporting high confidence, indicated by the two arrows. **b)** Usage of the preference and confidence rating scale with the two outliers removed. **c&d)** The same multiple regression analyses as presented in Figure 1 of the main manuscript, choice consistency and confidence are the dependent variables, total value, positive evidence and negative evidence are used as independent variables in different regressions, while controlling for value difference and variability. Data from all subjects are used in c and the outliers were removed in d. We find no major differences when we exclude the two outlier participants.

We have added this information in a new supplementary figure (new Supplementary Figure 1, page 19 in the revised manuscript).

6. *The finding that attentional effort is positively associated with confidence is exciting and novel (to me), and as the authors state - indicates that agents have some form of insight into the evidence accumulation process (since attention effort will reflect how balanced is their information seeking across options). It is particularly interesting because in the perceptual decision making literature attention towards a stimulus appears to lead to lower confidence judgements (e.g., Rahnev et al. 2011, Morales et al. 2015). Now, these two findings are not opposed, but it would be interesting to see the authors discussing their findings in this context too.*

Resp: This is indeed very interesting, thank you again for your input. We have incorporated these findings and literature to the discussion and how it relates to our findings (see page 11, right column, 1st paragraph of the revised manuscript).

“... This might be surprising in the light of results from the perceptual decision-making domain indicating that people overestimate their perceptual sensitivity for unattended stimuli and peripheral vision in general (Rahnev 2011, Kim 2005, Azzopardi 1973, Johnson 1986). This might lead us to believe that metacognitive performance is poor when peripheral vision is concerned and one could hypothesize that since the quality of peripheral vision is overestimated, confidence ratings based on peripheral vision are overestimated too. We have not explicitly tested for systematic overconfidence due to overestimation of the visibility of the unattended item. However, our results indicate that metacognition does correctly include information about peripheral vision in the sense that participants track the amount of effort they spend to incorporate the unattended item into the decision process and that more effort correlates with higher confidence ratings. ... “

7. *In the methods the authors write that trials where participants had not fixated on every option available at least once were excluded to ensure participants deliberated between options. Not fixating on one option might not necessarily mean that participants did not deliberate, because one can get the gist of a scene without focusing on each part. Please indicate how many trials were thus excluded for each participants. If the number is high, please replicate the main results using all trials (I think a recent study by Sepulveda et al. eLife 2020 where GLAM was used, all trials were included).*

Resp: We agree with the reviewer that one can get the gist for the whole visual space without changing fixation. Thus, as suggested by the reviewer we investigated the number of trials in which participants fixated only once. We found that for only 1.1% percent of trials this happened and therefore it is unlikely that including these trials may have affected our results. We have added a new supplementary figure (new Supplementary Figure 2, page 20 in the revised manuscript) where we show some fixation descriptive metrics, including the distribution in the number of fixations.

8. *There is a rich history in perceptual decision making that confidence reports do not follow a Bayesian optimal process - i.e., that confidence can be dissociated from accuracy or the strength of evidence (e.g., DePaulo et al 1997, Komura et al 2012, Koizumi et al 2015, Samaha et al. 2017, Maniscalco et al. 2021). It is true that in the animal literature studies have generally*

taken a Bayesian/statistical approach, but when reading the manuscript the impression was that in previous studies of confidence (esp. in perceptual decisions), confidence has been so far assumed Bayesian optimal and the authors' study was the first to show evidence against it.

Resp: The reviewer is right that this part of the text might be misleading in this sense. We have now made changes to clarify the rich history of studies investigating relations and dissociations between task accuracy and confidence and provided references to the corresponding articles (page 1, right column, paragraph 1 in the revised manuscript).

“Normative approaches formulate that confidence reflects an optimal estimate that the decision was correct (Pouget 2016, Meyniel 2015) and is a direct transformation of evidence strength (Kepecs 2008, Kiani 2009, Hebart 2016). This rational is particularly relevant to one of the central arguments that have sparked research in the last years: whether confidence represents an accurate index of decision uncertainty (Pouget 2016, Sanders 2016, Adler 2018). Interestingly, earlier studies found that people indeed rely on the strength of evidence, but do not directly consider the uncertainty on that evidence when they make confidence reports (griffin 1992), nor do they necessarily use the evidence available against their choice (Zylberberg 2012, Koizumi 2015). All these studies aimed at dissecting the relationship between choice behaviour and its resulting confidence evaluations (DePaulo 1997, Komura 2013, Koizumi 2015, Samaha 2017, Maniscalco 2021). However, this has not been formally studied in the domain of value-based choices. ...”

9. It was not immediately clear that the results in Fig3 were from the normative model.

Resp: We added a title in the figure indicating that the analyses and predictions are from the normative model and also added a clarification to the first sentence in the legend of figure 3c and also in panels 3g and 3h.

10. It seems Fig4a-b, 4e-f were not introduced in the text. Introducing the type of betting in the Folks task in Fig4e is kind of strange, since the data was already used in Fig3.

Resp: It is correct that we already use data of the Folke et al. study in Figure 3, so we have moved Figure 4e to Figure 3b, which allows to more easily compare the results obtained from our data collection and Folke's et al. study. We have also made sure that all figure panels are referenced in the main text.

11. There are many references to methods in the text that make understanding the paper/approach difficult at times (going back and forth).

Resp: We decided to avoid being too technical during the description of our modelling approaches in the main text, that is the reason of why we decided to provide higher level explanations in the main text and provide full details in the methods. Nonetheless, following the suggestion of the reviewer (and also reviewer #1), we tried to reduce references to the methods section and we opted to add a paragraph introducing the analyses and different modelling approaches to the reader before all modelling approaches are described in more detail (page 2, right column in revised manuscript). In this way, we hope that the reader finds it more digestible to follow the rationale of our modelling approaches throughout the article. We are happy to implement any specific suggestions that the reviewer and editor may have in this regard.

“Analyses and modelling roadmap

In the following we will present several analysis and modelling approaches, all with the goal of elucidating the influence of different components of the decision process on confidence reports. Here we briefly outline the different modelling approaches adopted in this work.

In the first part, we perform a "model-free" analysis investigating the influence of key decision variables on choice confidence reports, such as value difference and total value of the input alternatives, alongside the influence of value estimation variability obtained from the rating phases. The impact of these factors is separately studied in both choice consistency and confidence reports. These analyses will provide initial hints about the degree of similarity with which these decision variables impact choices and confidence.

In the second part, we study dynamical aspects of the decision process on confidence reports by jointly incorporating choices, reaction times, and fixation patterns. This will be formally studied based on sequential sampling models which will allow dissecting the influence of latent variables of the decision process such as the degree of attentional effort and the decision evidence gain on confidence reports.

In the third part, we investigate a modeling approach that does not make use of reaction time information to investigate the influence of the same latent variables investigated in the second part. This approach is appealing due to the following reasons: First, it is less computationally demanding. Second, access to reaction time data is not always possible in studies of perceptual and economic behaviour. Third, as we will show in the last part of this article, this approach can be parsimoniously extended to formally incorporate the statistics of the environment, which in turn allows to disentangle the influence of noise on confidence at different stages of the decision process such as the value encoding noise and downstream noise in the comparison process. ...”

12. Since eye-tracking data play a major role in the modeling, some basic / summary analysis / visualizations could be reported in supplementary.

Resp: Following the reviewer’s suggestion, we have included an additional supplementary figure reporting various summary statistics of the eye-tracking data (Supplementary Figure 2 page 20), see figure 3 below.

Fig. 3: Eye tracking data. **a)** Mean dwell times on the item presented up, down and of the chosen and unchosen item. Participants look longer at the item they choose and tend to look longer at the item presented at the top. In the whole figure error bars represent standard errors of the mean. **b)** Participants switch between looking at the upper and lower item. The number of fixations ranges between 1 and 9, all trials in which participants only fixated on one item were removed. This way 1.1% of trials were removed. **c)** Number of fixations relates to the absolute of the difference in value between the upper item and the lower item. **d)** Psychometric choice curve conditional on the location of the last fixation. While participants choose, they tend to look at the item of choice.

13. Section “Confidence reports are not related to trial-to-trial fluctuations in reward encoding”
While earlier on the term value was used, here the term reward is used. It might be obvious for some readers, but I feel it will be confusing to others (it was to me). I suggest using ‘value’ throughout since in this context reward feels intangible.

Resp: We agree that the term ‘reward’ on its own is confusing and have now avoided it throughout the manuscript. However, in some parts we feel it is appropriate to remind the reader that we are talking about values of items that are potential rewards to the participants. Only in those cases we have replaced ‘reward’ with ‘reward value’.

14. When comparing two correlations that imply a difference in effect this should be tested explicitly (fig 1d, though this is done elsewhere in the manuscript).

Resp: Following the reviewer’s suggestion, we have now added the test for correlation differences by directly comparing the posterior chains of the Bayesian correlation coefficient. Using the data of all participants, we found that difference between the correlation coefficients is marginally significant ($\Delta\rho = -0.39 \pm 0.24$, $P_{\text{MCMC}} = 0.049$). However, we found that removing the two participants that were outliers in the confidence reports (see response to comment #5 of this reviewer and

Supplementary Figure 1 for visualization and detection of the outliers) the difference between the correlation differences improved ($\Delta\rho = -0.53\pm 0.19$, $P_{\text{MCMC}}=0.02$). We report this result in the main text (page 3, left column, 2nd paragraph), we have also dedicated supplementary figure 3 to depict the result of this analysis, see figure 4 below.

Fig. 4: Effects of trial-to-trial variability on choice consistency and confidence at the participant level. This is a replication of Figure 1d of the article, however two outlier subjects who consistently report high confidence have been excluded (see Figure 3). Participant's average level of variability in the rating task had a negative influence on average choice consistency of that participant ($\beta = -0.56 \pm 0.17$, $P < 0.001$, $r = -0.55$), however this effect is not present for the same analyses performed on confidence reports ($\beta = -0.03 \pm 0.19$, $P = 0.44$, $r = -0.03$). The difference of the effect of average variability on choice consistency and confidence ratings is significant since 98% of the density of the posterior estimates is below zero ($\text{Diff. } \beta \text{ Var} = -0.53 \pm 0.25$, $P = 0.02$).

15. In figure 2e it would be useful if more information were given (e.g. in the legend): how are these regressions results obtained? What do bars and error bars represent?

Resp: Following the reviewer's suggestion, we have added more information to explain how the results of these regression was obtained. The results originate from two separate linear regressions, on the left: Confidence \sim Correct (Cor) + VD + TV + VD*Cor + TV*Cor and on the right Confidence \sim Cor + theta + RT + theta*Cor + theta*Cor*RT. We use two linear regressions to prevent problems with high correlations between explanatory variables and to separate the experimental input variables from the variables emerging from the choices and the generative models. Bars indicate the size of the standardized beta values and error bars the standard error resulting from the mixed effects models, stars indicate significant difference from 0.

16. The way figures are referenced in the text is not always clear. Some panels are not mentioned/introduced at all in the text.

Resp: We thank the reviewer for pointing this out to us. We have made references to all missing panels.

*** End of reviewers' comments *****

References cited in this letter

- Vickers, D. (1979) *Decision Processes in Visual Perception*, Academic Press.
- Hecce Castañón, S. et al. (2019) Human noise blindness drives suboptimal cognitive inference. *Nat. Commun.* 10, 1–11
- De Martino, B. et al. (2013) Confidence in value-based choice. *Nat. Neurosci.* 16, 105–110
- Polanía, R. et al. (2019) Efficient coding of subjective value. *Nat. Neurosci.* 22, 134–142
- Sanders, J.I. et al. (2016) Signatures of a Statistical Computation in the Human Sense of Confidence. *Neuron* 90, 499–506

REVIEWER COMMENTS

Reviewer #1 (Remarks to the Author):

The authors have done an excellent job in addressing mine and the other reviewers comments. I am happy to warmly recommend it for publication. This is an important paper and I hope that will spark a lively discussion in the field.

Reviewer #2 (Remarks to the Author):

I thank the authors for addressing all my comments and providing well-thought responses and/or rebuttals. The position of your paper is now stronger and the message clearer. I have no further questions/comments. Good work!